# A Review of Functional Separators for Lithium Metal Battery Applications

**DOI:** 10.3390/ma13204625

**Published:** 2020-10-16

**Authors:** Jooyoung Jang, Jiwoong Oh, Hyebin Jeong, Woosuk Kang, Changshin Jo

**Affiliations:** School of Chemical Engineering & Materials Science, Chung-Ang University (CAU), 84, Heukseok-ro, Dongjakgu, Seoul 06974, Korea; wndud2362@cau.ac.kr (J.J.); shanall@cau.ac.kr (J.O.); melon922@cau.ac.kr (H.J.); dntjr2943@cau.ac.kr (W.K.)

**Keywords:** lithium metal battery, separator, next-generation batteries

## Abstract

Lithium metal batteries are considered “rough diamonds” in electrochemical energy storage systems. Li-metal anodes have the versatile advantages of high theoretical capacity, low density, and low reaction potential, making them feasible candidates for next-generation battery applications. However, unsolved problems, such as dendritic growths, high reactivity of Li-metal, low Coulombic efficiency, and safety hazards, still exist and hamper the improvement of cell performance and reliability. The use of functional separators is one of the technologies that can contribute to solving these problems. Recently, functional separators have been actively studied and developed. In this paper, we summarize trends in the research on separators and predict future prospects.

## 1. Introduction

Rechargeable batteries are chemical energy storage systems that interconvert chemical energy and electrical energy through the redox reactions of cathode and anode materials. Fossil fuel-based energy cycles are being replaced by renewable energy cycles at a high pace. In this regard, energy storage devices have proven to be essential in fulfilling the global demands of sustainable energy supply and solving problems such as oil depletion and environmental pollution [1,2,3,4,5].

Electrochemical rechargeable batteries consist of two electrodes, the cathode and the anode, separated by an electrolyte and a separator. Their application varies widely from small electronic devices to electric vehicles. Thus, studies on the stability of these batteries are essential because it is directly related to human safety concerns. Additionally, the separator is one of the technologies that contribute to increasing the reliability of these batteries. In the commercial lithium nickel manganese cobalt oxide (NMC) battery cell (cathode: NMC 6:2:2|anode: graphite), the separator accounts for 7% of the price. Considering the fact that the global demand for separators is expected to be ~$1300 million in 2025 [6,7], it is essential to focus on the economic aspect of these batteries.

Generally, most commercial separators have porous structures that are fabricated using polyolefin polymers such as polyethylene (PE) and polypropylene (PP). The representative roles of the separator are as follows [8,9,10]: (1) it provides migration paths for ions but prevents electrons from flowing directly through the electrolyte. A high affinity (wettability) between the separator and electrolyte provides facile ion conduction, resulting in lower internal resistance. (2) It physically prevents direct contact between a cathode and an anode, which reduces the risk of short circuits or explosions. Therefore, mechanical strength is an important prerequisite to avoid shrinkage and rupture during cyclic process in different temperatures. (3) It should be sufficiently thin to enable the batteries to achieve short ion migration paths and high volumetric energy densities. The surface properties and porous structure of separators are important factors to be considered while designing advanced separators for various battery systems.

A recent research trend is the reviving of Li-metal batteries (LMBs), which use Li-metal as an anode owing to its high theoretical capacity (3860 mAh g^−1^), low density (0.59 g cm^−3^), and low reaction potential (−3.04 V vs. standard hydrogen electrode) [11,12,13]. In 1970, Whittingham developed a new battery system using Li-metal as the anode and TiS_2_ as the cathode [14]. In 1989, Moli Energy commercialized an LMB that used a MoS_2_-based cathode. However, because of the inherent instabilities of Li-metal caused by Li-dendrite formation, the product disappeared from the market due to the occurrence of safety accidents [15,16]. In 1991, Sony commercialized the modern Li-ion batteries (LIBs) by substituting Li-metal with graphite and using LiCoO_2_ (LCO) as the cathode material [17]. Thereafter, the optimization of LIBs resulted in the phasing out of LMBs [18]. Recently, as global demand for high-energy-density batteries has increased, LMBs are becoming popular again. In addition to its high energy density, Li-metal anode technology is important as next-generation batteries, such as Li-sulfur batteries (LSBs) and Li-air batteries, directly use Li-metal as anodes [19,20,21,22]. In particular, LSBs have attracted great attention owing to the high theoretical energy density and low cost. The use of sulfur as a cathode material accompanies a huge benefit in theoretical energy density (2600 Wh kg^−1^), providing opportunities to be used for electronic vehicles and portable electronic products [23,24].

However, for Li-metal to be commercialized, the following problems must be addressed: (1) during the Li plating/stripping processes, the volume of the Li-metal expands, which causes the solid electrolyte interphase (SEI) layer to crack. Repetitive SEI formation/decomposition decreases the energy efficiency and generates gaseous byproducts, increasing the pressure formation in the batteries. (2) During Li plating, the Li-ion flux increases through the cracks in the SEI layer which leads to the growth of non-homogeneous Li-dendrites. Dendrites grow rapidly and can penetrate the separator and reach the cathode, resulting in cell failure and explosion in the worst case [25]. (3) As the cycle continues, the growing dendrites are isolated, resulting in the formation of “dead” Li. The efficiency and capacity of the battery decrease when dead Li accumulates, and the SEI layer becomes excessively thick, disturbing ion transport. (4) These problems are severe in “harsh conditions” such as overcharging, elevated current densities, or fluctuations in temperature. (5) In the case of LSBs, intermediate polysulfide species (Li_2_S_x_, x = 8, 6, 4, and 3), formed during charge/discharge processes, are soluble in the electrolyte. These polysulfide species can diffuse and cover the anode surface, leading to performance degradation [26,27].

Various approaches have been adopted to stabilize and improve Li-metal anodes, such as changing either the morphology of the Li-metal or constituents of electrolytes [28,29,30], developing host materials with three-dimensional (3D) structures [31,32], and the use of current collectors [33,34,35]. Among these components, we focused on the separators. To the best of our knowledge, although several review articles on LMBs exist [36,37,38], a comprehensive review of separators for LMB applications has barely reported [39,40]. Since several papers have reported studies on improving the performance of Li-metal anodes by applying functional separators, this review examines the trends in the research on separators.

## 2. Properties of the Separator

The separator affects cell performance and safety. Therefore, understanding their characteristics and requirements for better performance is important. This section overviews commercial separators, specific parameters and analysis of LMB separators, and characteristics of separators for LMBs (Figure 1).

### 2.1. Thickness

Uniform thickness of the separator promotes homogeneous ion distribution, leading to the uniform use of the active materials present in the electrode layer and induces flat Li-metal formation by suppressing the growth of Li-dendrites [41]. Commercial separators have a thickness ranging between 20–25 µm [42]. Thin separators can maximize the energy density of batteries by providing more space for electrodes. However, thin separators increase the possibility of punctures and short circuits. In contrast, if a separator is too thick, it causes high resistance and decreases the energy density. In the case of LMB application, an optimal thickness should be systematically determined to prevent the growth of sharp Li-dendrites.

### 2.2. Porosity

Optimum porosity enables the electrolyte to be thoroughly wetted into the pores and provides facile ionic conduction. Generally, commercial separators with pores of 1 µm or less have a porosity of ~40% [42]. A high porosity reduces the mechanical strength of the separator and increases the possibility of punctures. In contrast, if the porosity is too low, electrolyte wettability decreases and the internal resistance increases. Therefore, an optimum thickness should be determined and the pores of the separator should have even morphology and pore size distribution. The homogeneous pore size distribution facilitates uniform ion distribution. Pore size should be sufficiently large to absorb the electrolyte and enable Li-ions to pass; however, it should be smaller than the size of the particles of the electrode material [41]. In the case of LMBs, a sub-micrometer pore size has proven to be adequate to block the penetrating Li-dendrites [43]. Generally, the porosity is measured by comparing the weight of the liquid electrolyte before and after absorption. Information on the surface topography and cross-section morphology can be obtained using a scanning electron microscope (SEM) [44]. In addition, nano-computed tomography (nano-CT) and mercury porosimetry techniques are applied to characterize porosity, pore structure/distribution, and pore sizes.

### 2.3. Wettability

The separator must absorb a sufficient quantity of electrolyte; during cell operation, the pores should retain the absorbed electrolyte [45]. If the wettability is high, the ionic resistance in the cell is lowered, which improves the cell performance. On the other hand, low wettability results in non-uniform ion distribution causing dendritic growth, causing electrode materials not to be fully used. Hence, wettability is crucial to both cell capacity and lifecycle. The degree of the wettability is indicated by the contact angle measured between the electrode and electrolyte. The contact angle is obtained by placing an electrolyte droplet on a dry separator and observing the droplet shape over time. A contact angle greater than 90° indicates poor wettability, and a lower contact angle indicates greater affinity between the separator and the electrolyte [44].

### 2.4. Ionic Conductivity

The ionic conductivity of a separator containing electrolytes is ideally expected to be in the range of 10^−3^ to 10^−1^ S cm^−1^ [44]. Generally, the MacMullin number is used to predict the ionic conductivity in the assembled cell. The MacMullin number is defined as the ratio of the resistance of the separator wetted with electrolyte to that of the electrolyte. The lower the MacMullin number, the better the cell performance and safety. Air permeability is used to estimate the MacMullin number, and it is expressed using the Gurley number and defined as the time required for air to pass through a specific region of the membrane under particular pressure. A low Gurley number indicates a high air permeability, high separator porosity, and low roughness. Generally, the separator requires a Gurley number lower than 25 s per 25.4 mm (ASTM D726) [44,45]. Chun et al. measured the air permeability of a commercial PP/PE/PP separator and carbon nanotube (CNT) separators with various isopropyl alcohol (IPA) volumes by measuring the Gurley value using a Gurley densometer (4110N). High IPA content in the CNT separators significantly decreased the Gurley value and MacMullin number and increased the ionic conductivity (Table 1) [46]. Ionic conductivity is calculated using the bulk resistance, which is obtained using electrochemical impedance spectroscopy, sample thickness, and area of the separator [44].

### 2.5. Chemical and Electrochemical Stability

The separator must be an electronic insulator [41]. Additionally, it should be electrochemically stable under redox reaction potentials. Using cyclic voltammetry or linear sweep voltammetry, electrochemical redox processes can be obtained to predict the stability of the separator [44]. For example, experiments were conducted to analyze the stability of a separator by measuring the cyclic performance using Li|Li symmetric cells [47]. In addition, chemical and electrochemical stability were analyzed by observing changes in the chemical state of the elements in the separator during the charge/discharge process using X-ray photoelectron microscopy and Fourier transform infrared spectroscopy [47].

### 2.6. Thermal Stability

The separator should be stably maintained over a wide temperature range [41]. The thermal stability of the separator is characterized using the melt integrity and the “shutdown function.” The melt integrity temperature is the temperature at which the separator can no longer maintain its mechanical properties, and the ideal melt integrity temperature is 200 °C or higher [44]. The separator begins melting down under abnormally high temperatures. Thus, it blocks the Li-ion conduction by eliminating the pores. This process is called the shutdown function. When it attains the shutdown temperature, the polymeric separator begins melting the pores and blocking the ionic flow. Mechanical integrity should be maintained to prevent contact between electrodes after shutdown [45]. Zhao et al. measured the shutdown performance of separators (Figure 2). They confirmed the shutdown process in PP/PE/PP and PE/Polyimide (PI)/S separators at 140 °C for 0.5 h and reported that the resistance increased after shutdown [48].

Ductile-to-brittle transition and melting temperature can be measured using thermogravimetric analysis and differential scanning calorimetry. The shutdown temperature and degree of shutdown can be quantified by measuring the impedance spectroscopy and temperature changes in the cell containing the separator [49,50].

### 2.7. (Thermal) Dimensional Stability

The shrinkage in commercial-grade separators should be less than 5% in all directions. The shrinkage is measured by comparing the areas of original separator and areas of separator which is impregnated with liquid electrolyte during a few hours. Also, the thermal shrinkage should be less than 5% after 60 min at 90 °C. [45]. Kang et al. reported the synthesis of a silica-PE separator and applied it in a graphite|LiMn_2_O_4_ cell (Figure 3). Silica prevents electrolyte/thermal shrinkage by mechanically maintaining the size of separator. Moreover, the pore structures achieved by binder-free, thin-layer growth of silica facilitate efficient ion transport. Consequently, the bioinspired silica-coated separator improved thermal stability (1 h at 140 °C) and exhibited high electrolyte wettability. The improvement of ionic conductivity would synergistically contribute to the enhanced rate capability of silica-PE separators compared with the unmodified PE and 2-dimethylaminoethanethiol(DMAET)-PE separators [51].

### 2.8. Mechanical Properties

The separator must have sufficient mechanical properties to withstand physical stress caused by external compression and electrode expansion. Generally, the mechanical properties of conventional separators differ when the separator is placed in an electrolyte. Therefore, mechanical stabilities in the electrolyte system must be considered to design high-stability separators. Mechanical properties are characterized by measuring tensile and puncture strengths; the higher the tensile strength of the separator, the better the rigidity. Also, a high puncture strength increases the resistance to dendrites, leading to prevent dendrite penetration. A separator should have a high mechanical strength to prevent dendritic growth in LMBs from penetrating it.

### 2.9. Preventing Shuttle Effects

In LSBs, cathode is known to have two plateaus during the discharge process. In the first plateau (~2.4 V vs. Li^+^/Li), sulfur is reduced from S_8_ to S_4_^2−^, at which various Li polysulfides (LiPS, Li_2_S_x_) dissolve in the electrolyte. In the second plateau (~1.95 V vs. Li^+^/Li), Li_2_S_4_ converts into insoluble Li_2_S_2_ and Li_2_S [52]. As-formed LiPS diffuses to anode side owing to electrostatic attraction between Li-metal and charged LiPS, and this phenomenon is designated as shuttle effects [53,54]. Shuttle effects lead to several degradations such as loss of active materials, low Coulombic efficiency (CE), and passivation of the Li anode [55]. Accordingly, separators have to prevent LiPS from migration. The chemical interaction between LiPS and separator can obstruct the diffusion of LiPS to anode side. In addition, the separator can enable the reuse of LiPS as active materials [56].

## 3. Limitations of Commercial Separators in Lithium Metal Batteries

Polyolefin is commonly used in commercial separators for LIB applications. In particular, PE and PP are mostly used, which are produced using wet and dry processes (Figure 4a–c).

Both processes include extrusion and stretching steps. Solvent extraction and annealing process are generally applied in the synthesis of these materials [8]. Polyolefin has high mechanical strength and is chemically stable and cost-effective. Additionally, its conductivity increases after it is soaked in an electrolyte because of its large pore volume and size. However, there are some inherent properties of polyolefin such as poor thermal stability and low wettability. The poor thermal stability can cause thermal shrinkage at high temperature, leading to short circuit. The low wettability can cause overpotential and capacity degradation at rapid charge/discharge rate [8]. Moreover, the affinity between the separator and a conventional polarized alkyl carbonate electrolyte solution is low because of the lack of polar groups in the separator. Therefore, the impeded ion transport and low compatibility of the interface hinder the migration of Li-ions, resulting capacity degradation. Also, uneven Li-ion distribution induces the growth of Li-dendrites, which can cause low cycling performance [60]. In particular, for polyolefin separators in LMBs, frameworks with large pores cause dendritic growth, resulting in poor stability.

Polyacrylonitrile (PAN), poly(methyl methacrylate) (PMMA), or poly(vinylidene fluoride) (PVDF) have been studied to overcome these drawbacks (Figure 4d,f). PAN-based separators can effectively suppress dendritic growth owing to their high modulus. Although they have high conductivity, good thermal stability, high electrolyte uptake, and good compatibility with Li-metal, PAN-based separators still suffer from the electrolyte leakage problem. PMMA-based separators have a high affinity and good compatibility with electrolytes, but their mechanical strength is weak because of their amorphous nature. The ionic conductivity of PVDF-based separators is higher than that of polyolefin materials owing to the superior wettability of PVDF-based separators. However, they cannot be applied to LMBs due to their weak mechanical strength and low thermal stability [61]. Therefore, novel strategies to modify commercial separator materials such as polyolefin are required. Several strategies to enhance the performance of separators have been suggested, and these include compositing other materials with polyolefin, introducing inorganic materials, and hybridizing organic/inorganic materials. These materials are discussed in the following section.

## 4. Various Materials for Modifying Multifunctional Separators

A commercial separator is primarily composed of organic materials such as PE or PP. Therefore, functionalizing organic materials on the separator is easy [50]. Compared with metals and ceramics, organic materials are light, cheap, and easy to apply to the commercial manufacturing process owing to their simple synthetic processes. Therefore, various forms of organic materials have been placed on commercial separators using relatively simple methods such as coating [62], vacuum filtration [63,64], and electron beam deposition [65]. A wide range of organic materials have their own characteristics such as ion conductivity [66], thermal stability [67], mechanical strength [68], and ability to suppress the shuttle effect in LSBs [69]. Owing to its extensive possibility of transformation, researchers have attempted to reinforce current commercial separators by applying organic materials [70,71].

On the other hand, inorganic materials have excellent physical strength. Thus, they have been used to increase the strength of separators and electrolytes or electrodes. In LIBs, silica coating is a typical method of improving the performance of the separator [72]. Additionally, a study on introducing inorganic and polymer hybrid layers as solid ion conductors in LMBs to suppress Li-dendrites was conducted [73]. Meanwhile, separators for LMBs should have additional competence to prevent the growth of Li-dendrites; in LSBs, depressing LiPS shuttle effects is required. Therefore, several researchers have introduced inorganic materials to separators to design functional inorganic separators. To date, several inorganic materials such as ceramic compounds, metal or metal oxide, nitrides, and phosphorus have been studied as additives or coating materials for separators.

### 4.1. Using the Advantages of Ceramic Compounds

#### 4.1.1. Silica (SiO_2_)

The reasons for using silica to improve the performance of separators are as follows: (1) Si is abundant and cost-effective since it is one of the most abundant elements on Earth [74]. (2) SiO_2_ has a high thermal stability; therefore, it can be used to increase the thermal stability of separators. (3) Because silica reacts with Li-metal [75], it can be used in separator surfaces to prevent direct contact with the cathode. (4) SiO_2_ has a high affinity to LiPS [76,77]; thus, it prevents the LiPS shuttle effect in LSBs. Therefore, the use of SiO_2_ in separators in LSBs is highly appropriate.

#### 4.1.2. Alumina (Al_2_O_3_)

Alumina has physical and chemical stabilities similar to silica and excellent LiPS adsorption [78]. Hou et al. clarified that the oxygen atom with the lone electron pair of ceramic compounds, such as alumina and silica, has a strong interaction with LiPS through dipole-dipole interactions [79]. Wang and co-workers fabricated a pure inorganic separator by the filtration of Al_2_O_3_ nanowires [80]. Generally, previous researchers focused on inorganic/polymer composite separators because of inorganic materials’ brittleness, which can cause cracking during cell operation. Indeed, inorganic materials are more beneficial in terms of thermal stability, electrolyte wettability, and overall mechanical properties than organic materials are. Hence, it is worth that this group successfully fabricated a pure inorganic separator by using the fine and uniform ceramic nanowire structure. Compared with commercial PP, this ceramic separator had enhanced porosity, liquid electrolyte uptake, ionic conductivity, and thermal stability. Additionally, this separator in a Li|LiFePO_4_ (LFP) cell exhibited better cyclic performance than that of a commercial separator. At high operation temperatures, batteries with Al_2_O_3_ separators exhibited stable cyclic performance, while the cell of a commercial separator made cell drop at first cycle.

### 4.2. Using the Advantages of Metal and Metal Composites

#### 4.2.1. Metal

Several studies attempted to stabilize LMBs by applying lithiophilic metal particles to separators. Metal particles are known to effectively suppress the growth of Li-dendrites for the following reasons: (1) lithiophilic metals such as Mg, Ag and Au effectively lower the Gibbs free energy required for Li electrodeposition, aiding to achieve uniform lithium deposition [81]. (2) Preparing porous and metallic substrates is easy; they act as Li host materials [82].

#### 4.2.2. Metal Oxide (MO)

Metal oxides can adsorb LiPS effectively due to the presence of oxygen atoms in the molecule; this reduces shuttle effects in LSBs [83]. For example, Xiong et al. coated a metallic oxide composite (NiCo_2_O_4_@rGO) onto a PP separator [84]. Transition metal oxides enable the catalytic conversion of LiPS during a recharging process because of their polar affinity. Moreover, NiCo_2_O_4_ exhibits a low energy barrier to the diffusion of Li-ions and provides abundant active sites for the catalytic conversion of LiPS (Figure 5). Therefore, it effectively prevents the accumulation of Li_2_S on the surface of Li-metal, increasing the stability of LSBs with a high areal capacity (7.1 mAh cm^−2^). Also, owing to MO’s high mechanical strength, it can stable Li-metal by preventing dendritic penetration.

#### 4.2.3. Transition Metal Dichalcogenide

A transition metal dichalcogenide (TMD) is a compound in which two chalcogenide elements are attached to a transition metal. The representative elements of a chalcogenide group are transition metal disulfides. TMDs have the following properties: (1) they have layered structures. (2) They provide channels for Li-ion intercalation [85]. (3) They have a high chemical affinity to LiPS. (4) They exhibit catalytic effects in the redox process of LiPS. Zhang et al. systematically revealed the correlation between the LiPS adsorption performance of 2D layered materials such as metal sulfides and the electrochemical performance of LSBs [76].

### 4.3. Using the Advantages of Carbon-Based Materials

#### 4.3.1. Carbon

Carbon has a complex shape and can form up to four stable bonds with other materials. Therefore, almost infinite types of carbon compounds are possible. In addition, porous carbons have high surface areas [86], high accommodation [87], and high ionic/electronic conductivities, making them very suitable materials for LMBs. Moreover, carbon is very light compared with other materials. By applying electrically conductive property, carbon materials can be loaded on the surface of separator to (1) change the direction of Li dendritic growth from carbon-coated separator to Li-metal and (2) promote the chemical reaction of LiPS dissolved from sulfur cathode, which occurs at the carbon-coated separator and sulfur cathode interface.

#### 4.3.2. Graphene

Graphene has a high electrochemically active surface area and a large mesopore volume, and is abundant. Therefore, it is commonly used to improve the performance of separators and other components in batteries. For example, Zhou et al. reported that a graphene coating on the one side of a separator and on the sulfur electrode offers rapid ion- and electron-transport, accommodates the volume expansion of sulfur, and suppresses shuttle effects by storing and reusing migrated LiPS [88]. Moreover, some studies introduced polymers such as polydopamine [89] or a transition metal oxide such as Li_4_Ti_5_O_12_ [90] together with graphene to improve the performance of LMBs. In all scenarios, cells using graphene-modified separators were demonstrated to have excellent electrochemical performance compared with that of commercial separators.

#### 4.3.3. Graphene Oxide (GO)

GO can be easily fabricated, is mechanically strong, and provides channels for ion diffusion owing to its 2D-layered structure. It is commonly used for rechargeable battery materials [83,91,92]. In particular, GO is used to enhance the performance of the separators in LSBs, of which the reasons were explained by Jiang et al. as follows: (1) GO contains abundant functional groups, which physically and chemically adhere to or entrap sulfur species. (2) It can effectively confine soluble polysulfide species into the porous skeleton, reducing shuttle effects and facilitating the continuous use of the polysulfide even in the long cycles. (3) GO enables a fast ion transportation and good rate performance since it has a porous structure. (4) The mechanical flexibility of GO can tolerate the volume expansion or shrinkage of sulfur cathodes; thus, it can maintain the integrity with cathode structures during charge-discharge processes [93]. Reduced graphene oxide (rGO) has been also applied in LSBs. The functional groups of rGO, such as epoxy and carboxyl groups, are beneficial to accommodating or preserving the sulfur species and increase the use of active materials [94].

### 4.4. Using the Advantages of Other Materials

#### 4.4.1. Nitrides (N)

Several attempts to apply nitrides, including boron nitride (BN), aluminum nitride, and niobium nitride (NbN), to separator applications have been conducted. BN is considered a crucial material in the field of energy storage devices because of its thermally stable, electrically insulating, and thermally conductive properties. Moreover, a BN-coated separator induces a conformal thermal distribution and stable SEI, resulting in uniform Li plating/stripping [95]. Hu group fabricated a separator by combining BN nanosheets with a poly(vinylidene fluoride-co-hexafluoropropylene) (PVDF-HFP) using a 3D printing technique [96] (Figure 6a). BN-separator improves the electrochemical performance of full cells by inhibiting Li dendritic growth. Paik group coated BN and carbon layers on the cathode and anode sides, respectively, of a commercial separator using an ink casting method for LSBs [97]. The carbon layer offered an additional electron transfer path and blocked dissolved LiPS. The BN layer protected Li from LiPS, controlling Li dendritic growth (Figure 6b).

In addition to BN, NbN were applied to improve separators. Lee group introduced flower-like mesoporous NbN to both sides of a commercial separator and applied it to an LSB [98] (Figure 6c). The high affinity between NbN and LiPS prevented LiPS from covering the Li anode and reactivates captured sulfur species, resulting in an increase in the reversible capacity. Moreover, the mechanical strength and electrolyte wettability increased, suppressing Li dendritic growth at the anode. Therefore, a full cell achieved low capacity decay (0.061% per cycle) during 300 cycles even at high sulfur loading (4 mg cm^−2^).

#### 4.4.2. Phosphorus (P)

Extensive research has been conducted to reinforce the properties of separators, using non-metallic and cost-effective phosphorus materials such as black-phosphorus (BP) and red-phosphorus (RP). Cui group synthesized BP nanoflakes to fabricate a bifunctional separator, which was assembled in an LSB [99]. BP is similar to graphite in appearance, structure, and properties: black, flaky, and a good electrical conductor (≈300 S m^−1^). The ionic diffusion coefficient of the phosphorene monolayer (zigzag direction) is 10^4^ times larger than graphene at room temperature. In addition, it can form chemical P-S bonds with LiPS and reactivate sulfur, resulting in higher specific capacity and cyclability even with a high sulfur content of 80%. RP is chemically stable, cheap, and easy to prepare and can chemically confine LiPS via Lewis acid-base interactions and sulfur chain catenation, which are appropriate for suppressing shuttle effects [100]. In addition, Li_3_PO_4_ was detected as a byproduct during the interaction between RP and LiPS, which promoted Li-ion conduction and aided in accelerating sulfur reaction kinetics. As a result, an LSB with the RP-based separator exhibited remarkable cyclability with a high capacity of 729 mAh g^−1^ after 500 cycles at 1 C (capacity retention of 82%).

## 5. Real Cases of Modifying Separators for Li-Metal Based Batteries

### 5.1. Separators for LMBs

#### 5.1.1. Strategies for Improving Ionic Conductivity of Separators

As introduced in Section 2, ion flux on the electrodes must be controlled uniformly to avoid Li-dendrite formation [101]. Therefore, researchers have coated functional materials with high electrolyte wettability to separators because of their ordered pore size and polar functional groups. Here are some papers that have given these properties to separator through materials coatings.

Choi group improved the performance of a commercial separator using polydopamine [102]. When polydopamine is coated on the PE separator, it changes the surface of the PE separator to be hydrophilic, increasing the extent of liquid electrolyte absorption. This enables Li-ions to be transported homogeneously to the Li-metal surface and inhibits the growth of dendrites (Figure 7a). The catechol moieties group, which is part of polydopamine, has excellent adhesion to versatile substrates. Moreover, since the bonding strength is excellent even in a liquid electrolyte, the separator can adhere well to the electrode. Since the separator adheres well to Li-metal electrodes, the deformation of the substrate during charge/discharge cycles could be minimized. After polydopamine coating, electrolyte uptake increased from 15 ± 2.7% to 112 ± 3.1%, ionic conductivity increased from 0.04 × 10^−3^ to 0.3 × 10^−3^ S cm^−1^, and 80% of the initial capacity was maintained even after 150 cycles in a Li|LCO cell. Sun group reported on PP separators coated with PVDF and Li_6.4_La_3_Zr_1.4_Ta_0.6_O_12_ (LLZTO) coated to increase ion conduction [103]. The interaction between PVDF and LLZTO creates a three-dimensional high-speed Li-ion channel along the PVDF/LLZTO interface. This can effectively transport and distribute Li-ions to the anode electrode, inhibiting dendrite growth. In addition, this separator immobilizes negative ions to evenly disperse Li-ions in the Li-metal surface.

In other studies, paper-type polymers were applied to increase ionic conductivity. Nyholm group prepared overoxidized polypyrrole (PPy) paper and cellulose composite films [104]. The overoxidizing process can electrically insulate PPy without structural damage (Figure 7b). In addition, both oxidized PPy and nanocellulose are hydrophilic properties, which increases the wettability of the electrolyte. Moreover, overoxidized PPy has higher thermal stability and ionic conductivity (1.1 mS cm^−1^) than commercial polyolefin-based separators. In Li|Li symmetric cell, this enables the cycle to operate for 600 h, thus proving to have a longer cyclic life than that of commercial separators.

There is a case of introducing ceramic material to increase ionic conductivity, taking advantage of entangled structure of polymer at the same time. Zhang and co-workers synthesized a separator using PAN and silica via centrifugal spinning. This cost-effective method developed separators with significant ionic conductivity and good wettability owing to the high porous fibril structure of PAN [107]. In this separator, PAN provided high ionic conductivity when the electrolyte was absorbed and had good thermal stability, with synergetic effects with SiO_2_. Electrolyte uptake was 310% and ionic conductivity was 3.6 × 10^−3^ S cm^−1^ in 12wt.% SiO_2_/PAN. They applied SiO_2_/PAN membranes to a Li|LFP full cell, which exhibited excellent rate performance with a capacity exceeding 160 mAh g^−1^.

#### 5.1.2. Strategies for Improving Mechanical Strength of Separators

The primary task of a separator is to prevent short circuits between the cathode and anode while maintaining ionic conductivity [82]. As described earlier, high mechanical strength is required to prevent dendrites from penetrating the separator [108]. Moreover, separators should have good electrolyte wettability and proper porosity [109]. In this section, high-modulus and porous materials coatings, which help in increasing the mechanical strength of separators, are discussed [61].

Ni group reported PVDF-HFP separator cross-linked with Al_2_O_3_ as the cross-linker [110]. The separator had a high ionic conductivity of 1.37 mS cm^−1^ in a Li|LFP half-cell. Because of the cross-linking and the presence of Al_2_O_3_, the mechanical strength was significantly increased to 30.4 MPa and thermal stability increased up to 180 °C. Kim group fabricated a high-strength separator using high-density polyethylene (HDPE) and ultra-high molecular weight polyethylene (UHMWPE) [105]. As the ratio of UHMWPE increased, the mechanical strength increased (Figure 7c). A film with 6wt.% of UHMWPE had a tensile strength of 1000 kg cm^−2^. In addition, it had uniform pores (0.1–0.12 µm) and excellent thermal stability that could withstand temperatures up to 160 °C.

Wang group fabricated an ultrastrong nanofiber membrane [106]. A nanoporous membrane was fabricated using a poly(p-phenylene benzobisoxazole) nanofiber (PBO-NF) through blade casting (Figure 7d). This separator was low cost and had a high strength of 525 MPa and Young’s modulus of 20 GPa. The membrane was stable up to 600 °C. In Li|Li symmetric cell, a pure Li-metal surface was observed after 700 cycles. It exhibited excellent performance in preventing dendrites growth. Kotov group synthesized aramid nanofibers (ANFs) [111]. In a layer by layer (LBL) structure, poly(ethylene oxide) (PEO) was applied in the ANFs as an ionic conductor. The tensile strength, Young’s modulus, and shear modulus were recorded as σ_ICM_ = 170 ± 5 MPa, E_ICM_ = 5.0 ± 0.05 GPa, and G_ICM_ = 1.8 ± 0.06 GPa, respectively. The crystallization of PEO, which is known to be detrimental to ion transport, can be controlled by the presence of ANF networks. As a result, in Li|Li symmetric cell, ionic conductivity was 1.7  × 10^−4^ S cm^−1^, which was higher than that of conventional polyolefin-based separators.

#### 5.1.3. Strategies for Improving Thermal Stability of Separators

For the commercial use of LMBs, thermal stability is a very important factor in separators. At high temperatures, the ionic conduction is very active, accelerating dendritic growth. Simultaneously, the separators lose their mechanical stability at high temperatures [112]. Polyolefin-based separators are not stable at high temperatures (130–160 °C) [48]. To complement this, a method of coating the separator with a ceramic-based substance has been developed. This can provide high thermal stability but has the disadvantage that ceramic materials can block the pores in the separator, complicating ionic transport and should use polymer binder such as PVDF-HFP [113] and PMMA [114]. In this section, we introduce studies that have improved thermal stability.

Lin group fabricated a sandwich-structured separator composed of PI/PVDF/PI using the electrospinning method [115]. This separator had a shutdown function (Figure 8). Because PI has a high thermal stability of 500 °C and low shrinkage, it is thermally and mechanically stable. The PVDF between the PI layers melted in 10 min at high temperatures above 170 °C. This is approximately 40 °C higher than that of a PE membrane. In addition, the electrolyte uptake was recorded at 476%, the ionic conductivity was 3.46 mS cm^−1^ in a Li|LiMnO_2_ coin cell, and the porosity was measured at 83%. Because of this, the battery had a high thermal stability, good cyclic life, and 95.1% capacity retention.

Li group fabricated polystyrene-poly(butyl acrylate) and silica(PS-b-PBA@SiO_2_) core-shell structures and used PS-b-PBA@SiO_2_ as a thermal shutdown switch [116]. The reason for covering silica is that it increases the thermal stability. PS-b-PBA@SiO_2_ was coated on a commercial PP separator to prepare a self-shutdown separator. PS-b-PBA@SiO_2_ fundamentally blocks the possibility of a short circuit of Li-dendrite because the porosity decreases to 7.5% at 80 °C. The separator produced using this method had almost the same electrochemical performance as a commercial PP separator in a Li|LFP cell.

Thermal stability can be enhanced by coating a nonflammable polymer on the separator. Xiong group synthesized a PAN and ammonium polyphosphate (APP) separator (PAN@APP) using electrospinning and applied this film as a separator in LSBs [117]. When the temperature increased rapidly, the PAN@APP could cross-link while releasing liquid and gaseous ammonia to create an insulating polymer layer. The characteristics and shape of the PAN@APP collapsed at 430 °C. In LSBs, the separator reduced the shuttle effect. The very strong interaction between LiPS and the separator, caused by the rich amine groups of APP and the phosphoric acid radical, prevented the LiPS from passing through the separator, exhibiting capacity retention higher than 83% for 800 cycles.

#### 5.1.4. Strategies for Stabilization of Li-Metal

Some methods to stabilize the Li-metal anode exist: (1) changing direction of Li dendritic growth by placing Li crystal seeds on the separator and (2) coating materials that react with Li-ions on the separator.

To address Li-dendrite formation, Liu group conducted a study on inhibiting dendritic growth by attaching polyacrylamide (PAM)-grafted GO to a commercial PP membrane (GO-g-PAM@PP) [91]. The GO-g-PAM@PP separator had high porosity that increased electrolyte uptake and provided Li-ion channels. Moreover, PAM has a strong lithiophilic property. Therefore, these properties guarantee rapid ionic conduction and low electrode/electrolyte interface resistance, resulting in stable cyclic performance. However, without the addition of GO, the PAM@PP film was brittle, breaking even at a small impact. By adding GO to PAM, the durability increased; it was no longer brittle even bent or folded conditions. As a result, a Li-metal electrode completed a 2600 h cycles at 2 mA cm^−2^ for a Li|Li symmetric cell, and a current density with high CE values (98%) for a Li|Cu cell was achieved.

Lee et al. used magnetron DC sputtering to coat a Cu thin film (CuTF) onto one side of a commercial PE separator [82]. The overall concept of this Janus-type separator is as follows: the side without CuTF is on the cathode side and remains as an insulator, while the CuTF side regulates Li dendritic growth at the anode side. The CuTF enables rapid electron conduction without interfering with ion transportation. Thus, the separator simultaneously enables facile electrochemical plating/stripping and inhibits the accumulation of dead Li. In addition, the deposited Li merges in the space between the CuTF and the Li-metal anode and expands along the surface of the anode; thus, internal short circuits are avoided (Figure 9a).

Song and co-workers introduced Mg nanoparticles on one side of a separator [81]. Lithiophilic Mg nanoparticles offer the sites for heterogeneous nucleation and produce a strong guiding effect to form fixed Li crystal seeds at the initial plating process, and consequently aid in retaining a dendritic-free and dense Li anode after the long cyclic process. Li nucleation occurs in separator-to-Cu direction rather than Cu-to-separator direction; this was confirmed using SEM after a Li|Cu half-cell test (Figure 9b). Cui group reported the silica nanoparticle sandwiched tri-layer separators by coating SiO_2_ nanoparticles between two commercial PE separators [118]. Previous studies focused on the use of SiO_2_ as a physical barrier because of its thermal stability and high wettability. This study emphasized the additional role of SiO_2_: it guides the growth direction of Li-dendrites due to the chemical reactions between SiO_2_ and Li. They conducted Li|Li symmetric cell tests after making pinholes on various types of separators to promote severe Li growth conditions to investigate the formation mechanisms. Four types of separators (bare, SiO_2_ coated, Si coated, PMMA coated) were used for the experiments. The SiO_2_-coated separator exhibited the longest lifespan (≈152 h) among the others (Figure 9c).

Yuan group fabricated a ZrO_2_/polyhedral oligomeric silsesquioxane multilayer-assembled PE separator, which was synthesized using a simple LBL self-assembly process [119]. This separator effectively reduces electrolyte polarization and protects Li-metal anodes from Li dendritic growth, and it exhibits excellent electrochemical performance and stability. Xie group reported an interesting strategy, which guided the direction of dendrite growth [101]. Their concept was to allow dendritic growth from both separator and Li-metal surfaces. These Li layers grew by facing each other, resulting in a fused and dense Li formation. This concept was realized by coating a conductive carbon layer on the separator surface, which faced the Li-metal anode. This structure enabled dendrites to spread widely in a direction parallel to the electrode. The Li-metal electrode exhibited a stable cyclic life with a capacity retention of 80% even after 800 cycles. 

#### 5.1.5. Separators Made from Non-Toxic and Sustainable Processes

Using toxic materials during the process of fabricating separators can cause environmental problems. For this reason, several studies on using non-toxic water-based solvents to be eco-friendly and reduce cost have been conducted.

Lee group created a concept of a plasma-treated ceramic-coated separator (plasma CCS) [120]. In contrast to previous studies, in which toxic organic solvents were applied, they used cost-effective and eco-friendly water-based plasma treatment to modify the surface of PE separator, increase the pore size, and strongly fix the alumina layer. Additionally, a ceramic-coated separator was prepared using the surfactant technique (surfactant CCS), which was commonly used in previous studies. A Li|LiMn_2_O_4_ cell using plasma CCS exhibited better performance than other batteries in terms of discharge capacity, resistance, and cyclic performance.

Peng et al. introduced graphene to a separator. They coated cellular graphene frameworks onto a PP separator, suppressing the migration of LiPS [121]. However, used synthetic methods, chemical vapor deposition (CVD) with low yields and vacuum filtration processes are not cost-effective for mass production. Hence, they developed Janus-type porous-graphene (PG) modified separators, which were scalable and suitable for green fabrication [60]. First, they used fluidized-bed CVD, which had a yield of 5 g h^−1^. Additionally, an industrially compatible blade coating method was applied, and toxic organic solvents were replaced with water. An amphiphilic polymer with a high polarity, poly(vinyl pyrrolidone), was adopted as the aqueous binder because of its wettability to the PP separator and PG and polarity for LiPS interaction. Consequently, the PG separator induced a significantly low self-discharge rate (90% retention) at high sulfur use (86.5%) and increased the rate capability.

In addition, Lei et al. also fabricated separators using Al_2_O_3_ nanowires under mild conditions [122]. The existing methods to synthesize Al_2_O_3_ nanowires cannot be easily scaled up, since the process is complicated and requires harsh solvents. In their study, they introduced a facile extraction process to extract Li from Al or Mg alloys using alcohol solvents for the synthesis of alumina nanowires. The as-made separator exhibited increased thermal stability, ionic conductivity, and wettability compared with commercial PP and cellulose fiber separators. When applied to a graphite|LFP cell, it exhibited much better performance than other batteries in terms of capacity retention, rate capability, etc. This research created the possibility of the mass production and commercialization of ceramic separators in which Al_2_O_3_ nanowires were synthesized and applied as a separator under ambient conditions without a catalyst or external stimulus.

Wu et al. reported a functional separator by stacking a Prussian blue (PB) layer on an rGO film (PB/G) [123]. PB, a type of MOF, is stable, non-toxic, and scalable material and has an appropriate lattice size and open framework with large interstitial sites. Therefore, it can accommodate Li-ions while obstructing the migration of LiPS [124]. The PB barriers are evenly distributed at the anode side and alleviate the growth of Li-dendrites by maintaining a homogeneous Li-ion concentration. PB/G is on the cathode side, hindering the diffusion of LiPS and increasing the conductivity of a cell. With these characteristics, a Li|S full cell achieved a high capacity of 1481 mAh g^−1^ at 0.1 C and 744 mAh g^−1^ after 2000 cycles at 2 C.

Kim et al. have developed an eco-friendly coating process of GO to fabricate modified functional separators [125]. Through this method, surface of the separator can be fully covered by GO flakes, which provided the hydrophilic wetting nature owing to many hydrophilic functional groups existing on the GO, enabling eco-friendly water-based slurry method. They fabricated GO-SiO_2_ composite layer coated separator to confirm its applicability to LMBs. The SiO_2_ nanoparticles performed the function of suppressing Li dendritic growth, and exhibited more stable cycling performance compared to bare separator. Also, this study emphasized that this method can be used with not only SiO_2_ but also other 1- or 2-dimensional materials including CNT, graphene, TMD, etc.

### 5.2. Separators for LSBs

As described earlier, currently used polyolefin-based separators cannot prevent LiPS crossover during the LSB cycles. Therefore, several studies on blocking LiPS in a separator are being conducted [126]. Establishing a multi-functional separator that increases the conductivity of the Li-ions and effectively blocks LiPS is necessary. Electrostatic repulsion and chemical trapping of LiPS are actively investigated approaches to achieve this aim. In this section, we introduce studies on suppressing the shuttle effect chemically and physically using various materials.

#### 5.2.1. Strategies for Suppressing Shuttle Effects by Chemical Methods

Polymers have an advantage in that they can be prepared in multiple layers. Therefore, a separator that exhibits the synergistic properties of each polymer can be manufactured. For example, Lee group used multi-walled carbon nanotube (MWCNT)-wrapped polyetherimide nanomats as top/bottom layers and poly(1-ethyl-3-methylimidazolium) bis(trifluoromethanesulfonyl)imide (PVIm[TFSI])/poly(vinylidene fluoride-co-hexafluoropropylene) (PVDF-HFP) as a middle layer [127] (Figure 10a). In this structure, TFSI ions chemically trapped LiPS, alleviating the shuttle effect. In addition, PVIm[TFSI] in the middle layer improved the mechanical properties as a pillar and the PEI/MWCNT in the top/bottom layers acted as a current collector. These multiple layers promoted the redox reaction of the cathode but prevented the shuttle effect. Xu group conducted a study on blocking LiPS using functional groups on the separators [128]. They prepared a gum arabic (GA) and conductive carbon nanofiber (CNF) mixture. GA is composed of branched polysaccharides, which are composed of alactose, ramnose, arabinose, and hydroxyproline side chains. Thus, it traps LiPS using hydroxyl, carboxyl, and ether functional groups. The LSB with this separator exhibited excellent performance: 94% capacity retention even after 250 cycles.

Various inorganic materials which have functional groups with great affinity to LiPS also can take a significant role of suppressing shuttle effects. Wang group fabricated a PP-SiO_2_ separator by simply immersing a PP separator in the hydrolysis solution of tetraethyl orthosilicate with the assistance of Tween-80. When this separator was applied to an LSB, cyclic stability and rate capability increased considerably because the PP-SiO_2_ separator (1) enabled facile Li-ion transportation owing to its improved wettability, and (2) mitigated the shuttle effect owing to a strong physicochemical interaction between SiO_2_ and LiPS (Figure 10b). Lai et al. applied a hydrothermal method to fabricate hollow carbon nanofiber@mesoporous δ-MnO_2_ nanosheets (HCNF@pδ-MnO_2_). This nanomaterial was coated on the commercial separator [129]. Birnessite-type MnO_2_ (δ-MnO_2_) exhibits a strong chemical affinity to LiPS through surface bonding and is considered an effective sulfur host. The δ-MnO_2_ prevents long-chain LiPS from dissolving into the electrolyte and promotes the deposition of short-chain LiPS; thus, it is suitable for modifying the properties of separators in LSBs. However, since δ-MnO_2_ has an intrinsically low electrical conductivity, HCNF was selected as the electrically conductive material. Additionally, HCNF can form a robust 3D matrix because of its high length/diameter ratio and can be easily combined with δ-MnO_2_. As a result, the separator can suppress shuttle effects physically and chemically, resulting in stable cyclic stability (Figure 10c).

Xiang et al. improved the performance of LSB by applying SnO_2_ to commercial separators [131]. This separator increases the mobility of Li-ions by increasing wettability and inhibits the migration of LiPS by strong physical and chemical interactions between SnO_2_ and LiPS. Furthermore, since SnO_2_ coating slightly affects the mass or volume of the separator, a high energy density of an LSB can be maintained. Kang group fabricated a sandwich composite interlayer, placing a VS_2_/CNT composite on a CNF framework and covering graphene on it [132]. VS_2_/CNT effectively reduces the self-discharge phenomenon of LSBs owing to its strong affinity to LiPS. Specifically, VS_2_ has a function of forming polar V-S groups with LiPS, while the CNT forms a 3D conductive network. CNF substrates synthesized on PAN as a supporting framework increase the wettability and facilitate Li-ion conduction. The graphene coating layer serves as a second current collector and effectively recovers the inactivated sulfur species. An LSB with this separator achieved a high capacity (1150 mAh g^−1^ and 750 mAh g^−1^ at 0.1 C and 0.3 C, respectively) even at high sulfur loading (5.6 mg cm^−2^).

Moreover, Kim group fabricated a SnS_2_-modified separator to facilitate the redox reaction of LiPS [133]. SnS_2_ is a conductive, polar, and catalytic material used in the conversion reaction of LiPS. Therefore, a SnS_2_ nanosheet-modified separator stabilized LSBs by capturing LiPS, facilitating ion diffusion, and performing additional current collector functions. Additionally, Tang group reported a MoS_2_/Celgard composite separator that acted as an effective LiPS barrier in LSB [134]. 2D flexibility and high Li conductive property of MoS_2_ were effective in suppressing LiPS shuttle effects. Furthermore, nitrogen-doped carbons have been actively investigated in LSB separator applications [135]. Nitrogen-doped carbon has high ionic conductivity and chemically absorption ability. Zheng group coated 2D porous nitrogen-doped carbon nanosheets on a commercial PP membrane [130] (Figure 10d). The density and thickness of the coated layer were 0.075 mg cm^−2^ and 0.9 µm, respectively. The high surface area of this layer trapped LiPS. In addition, since the materials gathered compactly, an excellent barrier effect was achieved even with a small weight. After nitrogen doping, the chemical interaction between LiPS and the carbon layer increased, and the film also acted as an extended current collector, enabling an 88.6% capacity retention after 500 cycles.

#### 5.2.2. Strategies for Suppressing Shuttle Effects by Physical Methods

Some functional separators can block LiPS by physical phenomena including electrostatic repulsion and structural trap. In this subsection, various cases which prevented LiPS from migrating through separator will be discussed. Ding group fabricated a PP separator coated by graft poly(acrylic acid) (PPA) for LSB applications [136]. PP grafted with PPA (PP-g-PPA separator) selectively passes Li-ions well, but LiPS is blocked by electrostatic repulsion. Thus, LiPS did not pass toward the anode and remained in the separator at the cathode side. In addition, because of the strong bonding energy of PP and PPA, consistent and uniform blocking of LiPS during the long cyclic process was observed. The LSB exhibited an initial capacity of 800 mAh g^−1^, and 580 mAh g^−1^ after 250 cycles. Yamauchi group reported a double-layered modified separator as a shuttle suppressing interlayer for LSBs [137]. They fabricated this double-layered separator combining a microporous PP matrix layer and an arrayed PMMA microsphere retarding layer. In this separator, the arrayed PMMA microspheres blocked the diffusion of polysulfides chemically and physically. Ester groups of PMMA interact with polysulfides and chemically adsorb them. Moreover, the arrayed PMMA microspheres are easily formed by self-assembly. Owing to its physical and chemical effects on suppressing diffusion of polysulfides, a sulfur with the PP/PMMA separator exhibited an initial capacity of 1100 mAh g^−1^ at a current density of 0.1 mA cm^−2^, which is higher than the first discharge capacity of the commercial PP separator (948 mAh g^−1^).

Zhang et al. prepared an Al_2_O_3_-modified separator for LSBs and suppressed the shuttle effects effectively [138]. In their study, an Al_2_O_3_ layer was coated onto a commercial separator using slurry coating. In this separator, well-connected voids existed between nanoparticles, aiding Li-ions to move freely. Additionally, the entangled structure reduced the shuttle effects. Choi group used permanent dipoles in BaTiO_3_ (BTO) particles to effectively prevent LiPS from passing through a separator by electrostatic repulsion [139] (Figure 11a). A BTO coating increases the mechanical strength of a PE separator to prevent thermal shrinkage. Consequently, the BTO-coated separator in a Li|S full cell exhibited a high specific capacity and extended cyclic life.

Cui group coated a 30-nm carbon nanoparticle and polyvinylidene mixture (9:1 in mass ratio) on one side of a PP separator. This separator enabled a large quantity of LiPS to be accommodated in the separator layer [26]. Therefore, a large quantity of LiPS could be located on the cathode side of the separator. During 500 cycles, the LSB exhibited an initial specific capacity of 1350 mAh g^−1^ at 0.5 C and achieved stable cyclic performance with a decrease of 0.09% in every cycle. Giebeler group coated mesoporous carbon on a PP separator to trap LiPS to be placed in the cathode [140] (Figure 11b). Goodenough group fabricated a rGO@sodium lignosulfonate (SL)/PP separator using a thin coating of rGO/SL on a standard PP separator [141] (Figure 12). They used the principle that a negatively charged separator effectively suppresses negatively charged LiPS ions. The SL, a byproduct of chemical industries, contains an abundant quantity of negatively charged sulfonic and dendritic groups; thus, it was used to inhibit the shuttle effects. The flexible characteristic of rGO successfully prevented SL from flaking off from PP. It functioned as a robust separator, which guaranteed fast ion transportation, and delayed the migration of LiPS.

An MOF is a composite of metal ions and organics. It has a very large surface area and highly ordered pores, and it has been studied as a cathode host material for LSBs. However, the framework gradually degrades and its capacity decreases owing to its naturally insulating property. However, MOFs can selectively pass and trap ions because of their precisely adjustable pore sizes. Therefore, researchers have applied MOFs to block LiPS while facilitating Li-ion conduction by applying their porous structures and insulating property [92].

MOFs are frequently used in the coating process with GO or carbon nanotube (CNT) because of their weak mechanical and poor conductive properties. Zhou group conducted a study to improve the performance of a separator by combining a MOF and GO [92]. The parallel GO layer was coated on the uniform crystalline MOF nanoparticles. Cu_3_(BTC)_2_ (HKUST-1) was used as the MOF. Its 9Å pore size was smaller than that of LiPS. Therefore, it could decrease the shuttle effect. In their study, the poor cyclic life of the MOF separator exhibited an increasing LiPS crossover during the cycles. However, the GO in MOF@GO acted as a barrier and increased Li-ion conduction. The MOF@GO-separator LSB exhibited an initial capacity of 1126 mAh g^−1^ and a capacity reduction of 0.019% per cycle for 1500 cycles.

Chen group prepared CNT and MOF separators [142]. A zeolitic imidazolate framework (ZIF) was used as MOF and combined with MWCNTs (CNT@ZIF). The nano-sized MOF particles were not evenly dispersed and tended to gather (Figure 13). Such agglomeration causes non-uniform coating, which causes LiPS to pass through the separator. Therefore, Zn ions were added to MWCNTs to evenly disperse them. Additionally, CNT@ZIF composites were coated onto one side of a PP membrane. Because MWCNTs are electron conductors, coating the separator film is a simple approach to efficiently use the sulfur cathode. However, if MWCNTs penetrate the separator, a short circuit can be induced owing to the electronic conductivity of carbon. Additionally, by adding MWCNTs, a separator that can have both ionic conductivity and mechanical strength can be obtained. Therefore, a high capacity of 1588 mAh g^−1^ can be obtained at 0.2 C, and a 36.2% higher capacity retention can be obtained in LSBs compared with batteries with commercial separators for the first 100 cycles. Park group synthesized a functional separator by applying the vacuum filtering of MOF and Nafion on a PE membrane [143]. Zr_6_O_4_(OH)_4_(BDC)_6_ (BDC = 1,4-benzenedicarboxylate), termed as UiO-66, was used as the MOF. A sulfonic acid functional group (-SO_3_H) was attached to the MOF. Since UiO-66 has a smaller pore size than LiPS, it can block LiPS and -SO_3_H can prevent LiPS diffusion through electrostatic repulsion. Here, Nafion acts as a framework of the separator and provides a selective ion channel.

## 6. Hybridized Use of Electrolytes and Separators: Solid and Gel Polymer Electrolytes

In commercial LIBs, organic solutions containing Li salt have been used as liquid electrolytes. However, flammability, leakages, and decomposition problems have caused cell expansion and severe safety problems [144]. To overcome these disadvantages, several studies have been conducted to develop robust solid electrolyte (SE) and gel polymer electrolyte (GPE) materials. The limitations of commercial liquid electrolyte-based LMBs, such as dendritic growth and continuous electrolyte decomposition reactions, can also be addressed by applying these SEs or GPEs that can simultaneously function as separators and electrolytes [145,146,147].

Although these electrolytes are considered promising, persistent problems still exist: both electrolytes have limited kinetic properties because of their low conductivity at room temperature and high interfacial resistance [148]. GPEs have suffered from a poor mechanical strength and low CE values, while SEs suffer from Li dendritic formation and severe chemical reactivity at the SE/Li-metal interfaces. These problems can cause side reaction, safety problems and poor life-spans [149,150]. In this section, we introduce studies on GPEs and SEs that aimed to stabilize LMBs by suppressing the growth of Li-dendrites using only an electrolyte without conventional polymer separators.

### 6.1. Interfacial Resistance and Instability Resulting in Low Capacities

The major causes of SE failure are the decomposition of electrolytes by electrochemical and interfacial instability, volume change during the cyclic process, and short circuit owing to dendritic growth [151]. The deficient contact between SEs and the Li-metal anode results in non-uniform current distribution and facilitates the growth of dendrites. In particular, solid inorganic electrolytes cannot withstand the volume change during charge/discharge processes, resulting in cracks or defects. Moreover, interfacial reactions occur between the anode and electrolytes when the cathodic limit of the SE is lower than the electrochemical potential of the anode materials. This reaction forms an interfacial layer, which significantly increases the interfacial resistance [152].

PEO-based derivatives have been widely used as host materials in GPE fabrication. This is because their ether chains have strong interactions with Li-ions and electrolyte solvents. Nevertheless, the as-made GPEs cause short circuit problems, decreasing the mechanical strength through plasticization induced by organic solvents. Typical cross-linking reactions are frequently initiated by thermal radicals, such as benzoyl peroxide, di(4-t-butylcyclohexyl) peroxycarbonate, and azobisisobutyronitrile. This causes a drawback in that residual monomers and thermal initiators such as free radicals are very reactive with Li-metal. As these byproducts cover the Li-metal surface, the resistance increases, and the performance of batteries deteriorates. Fabricating GPEs not involved with thermal initiation is crucial to overcoming this disadvantage [148].

### 6.2. Modified SEs

#### 6.2.1. Composite SEs

To solve the challenge of interfacial resistance in SEs, Wang et al. suggested a method of coating nanoscale and lithiophilic zinc oxide (ZnO) onto the surface of a garnet-type SE [153]. Generally, the garnet-type SE is widely used in SEs owing to its high energy density, electrochemical stability, high temperature stability, and safety [154]. Because of the ultrathin and conformal ZnO surface coating, molten Li reacts with ZnO and produces better contact with the surface of the garnet electrolyte by reducing the interfacial resistance (Figure 14a). Although the conformal ZnO layer can be coated on the internal structure of 3D porous garnet SE, some methods such as sputtering or CVD cannot achieve the uniform coating. Therefore, the atomic layer deposition technique was successfully applied, which provided a good infiltration of Li-metal into the porous garnet by increasing the wettability of 3D porous garnet SEs. This garnet electrolyte coated with 30 nm of ZnO lowered the interfacial resistance to 20 Ω cm^2^, which was significantly lower than that of an untreated sample (~2000 Ω cm^2^), and maintained the stability during the plating/stripping process.

Hu and co-workers attempted to lower the interfacial resistance by depositing a thin germanium (Ge) layer onto a garnet electrolyte to provide a better contact between the Li-metal anode and the garnet [155] (Figure 14b). To fabricate the garnet pellet, they used LiOH, La_2_O_3_, and ZrO_2_ as starting materials, and SE was fabricated with the composition of Li_6.85_La_2.9_Ca_0.1_Zr_1.75_Nb_0.25_O_12_ (LLZO) with the addition of CaCO_3_ and Nb_2_O_5_ to lower the manufacturing temperature. Ge was coated onto the LLZO SE, and a Li-Ge alloy was formed to be used as a Li-ion conductor between Li-metal and the garnet. The Li|Ge-modified-garnet|Li cell exhibited an interfacial resistance of 115 Ω cm^2^. In terms of cycling stability, a Li|Ge-modified-garnet|LFP full cell exhibited capacity of 140 mAh g^−1^ up to 100 cycles with an efficiency of ≈100% at 1 C.

Research on novel SE has attracted interest, even in the field of LSBs. Fu et al. fabricated a 3D bilayer garnet SE framework [156]. One side of the garnet SE layer had a thick and porous structure, while the other side was thin and dense. The dense layer suppressed dendritic growth and functioned as a rigid barrier. Simultaneously, the porous layer supported the dense layer mechanically. This solid-state framework locally confined cathode materials and tolerated the volume change of the materials such as solid sulfur and polysulfide catholytes. The extent of sulfur cathode loading of an LSB went up to > 7 mg cm^−2^, exhibiting an initial CE of 99.8% and an average CE of 99%.

#### 6.2.2. Artificial Interface Layers between Li-Metal Anodes and Electrolytes

An artificial SEI layer between a Li-metal anode and electrolyte can suppress Li dendritic growth, which is a function of the separator. Fan et al. demonstrated that a fluorinated SEI layer between a Li-metal anode and LiPS_4_ (LPS) SE restrains the formation of Li-dendrites and prevents side reactions [157]. An LiF-rich SEI is easily formed by contacting LiFSI-coated/infiltrated LPS onto the Li-metal. An artificial SEI has high interface energy with Li-metal and a strong modulus; thus, it has an important function of suppressing the growth of dendrites. When a symmetric Li|LiFSI@LPS|Li cell was fabricated (Figure 15a), the CE increased from ~88% to ~98% compared with a Li|LPS|Li cell. Moreover, the critical current density increased from 0.7 mA cm^−2^ to a record-high value of >2 mA cm^−2^.

As another example, Fan and co-workers deposited a soft polymer electrolyte (SPE) layer on the surface of a garnet electrolyte (Ta-doped LLZO (LLZTO)) to increase interfacial conductivity, decrease polarization, and increase CE [152]. Furthermore, they fabricated a 3D Li-metal anode using a melt infusion strategy to improve the interfacial interactions. This strategy was demonstrated in their previous study [158]. An artificial SPE layer supports the mechanical role of separators by inducing the better contact between the electrode and electrolyte, suppressing dendritic growth. With the effect of increased wettability from the SPE-coating layer, an LFP|SPE-LLZTO-SPE|3D Li full cell (Figure 15b) at 90 °C exhibited a CE of 99.6% and a stable cyclic life of 135 mAh g^−1^ after 200 cycles.

### 6.3. Modified GPEs: Strategies to Enhance the Function of Separators

Although several studies on PEO, PAN, and PMMA have been conducted, applying them is difficult since their ionic conductivities at low temperatures are too low [159,160]. As a result, GPEs have attracted much interest. When GPEs are cross-linked, liquid components are incorporated into a polymer matrix; thus, they block the leakage of liquid electrolytes and induce high ionic conductivity. However, GPEs can be easily penetrated by Li-dendrites because of their low mechanical strength. To overcome this, several approaches have been applied to ensure stability against dendrite growth by increasing the mechanical strength using additive materials. In addition, strategies to increase ionic conduction have been introduced.

Various strategies for designing separators with the use of GPEs exist: (1) introducing inorganic materials that have strong mechanical properties to effectively suppress dendritic growth, (2) designing specialized structures to enhance ionic conductivity and mechanical properties, and (3) forming an electrolyte interphase layer between Li-metal anodes and electrolytes.

To suppress dendritic growth by using inorganic materials to increase the mechanical properties, Zhou et al. designed a hollow SiO_2_ nanosphere-based composite SE (SiSE) [161]. This hierarchical SiO_2_/polymer composite electrolyte was fabricated using the in situ polymerization of tripropylene glycol diacrylate (TPGDA) (Figure 16). To overcome the poor contact resistance between the electrodes and SE, they integrated SiSE and a TPGDA-based GPE. The cross-linked TPGDA polymer framework protected the SiSE to maintain a safe quasi-solid-state, decreasing the risks of electrolyte leakage. This electrolyte was liquid, absorbed in a hollow SiO_2_ nanosphere layer, providing both high ionic conductivity and low interfacial resistance. Several electrochemical tests depicted the performance of an LFP|SiSE|Li cell. This cell exhibited an ionic conductivity of 1.74 × 10^−3^ S cm^−1^ and low interfacial resistance. Moreover, it had an ion transference number of 0.44 and stable voltage window up to 4.91 V. The discharge capacity and CE were 158 mAh g^−1^ and 98.6%, respectively. After 200 cycles at 0.2 C, it had a discharge capacity of 159 mAh g^−1^ and exhibited a capacity retention ratio of 100.2%. This result implied that the quasi-solid-state strategy of combining GPE and SE can provide a synergetic effect that eliminates the drawbacks of each.

In the GPE field, existing cross-linking reactions have obstacles created from thermal initiation caused by free radicals and residual monomers. To avoid these disadvantages, Yang and co-workers reported a novel initiator-free one-pot synthesis strategy to fabricate a tough and compact 3D network GPE using a ring-opening polymerization reaction [148] (Figure 17a,c). They used the diglycidyl ether of bisphenol-A (DEBA) as a supporting framework to increase the mechanical strength. Meanwhile, poly(ethylene glycol) diglycidyl ether and diamino-poly(propylene oxide) were cross-linked through this framework for fast ion transport. The compact structure of the 3D-GPE forced the SEI layer to be formed homogeneously, obstructing the growth of dendrites. The as-made 3D-GPE successfully suppressed the growth of dendrites, created a stable interface, and complemented the ionic conductivity. Therefore, a 3D-GPE-based battery exhibited better electrochemical performance than a battery using liquid electrolytes and typical separators.

Similarly, Li and co-workers designed a novel dual-salt lithium bis(trifluoromethanesulfonyl)imide–lithium hexafluorophosphate (LiTFSI-LiPF_6_) GPE with a 3D cross-linked polymer network [162] (Figure 17b). The 3D cross-linked polymer network by poly(ethylene glycol) diacrylate (PEGDA) and ethoxylated trimethylolpropane triacrylate was formed using dual-salts. Accordingly, higher thermal stability and ion transference were obtained. The compact GPE solved contact problems by facilitating the uniform deposition of Li atoms and successfully restricted the growth of dendrites. Moreover, the linear chain motion of PEGDA increased ionic conduction. As a result, the measured ionic conductivity at 25 °C was 5.6 × 10^−4^ S cm^−1^, which was higher than that of single salt GPE, LiTFSI (1.6 × 10^−4^ S cm^−1^) and LiPF_6_ (1.2 × 10^−4^ S cm^−1^).

A remarkable method of converting liquid electrolytes to quasi-solid GPEs by the addition of commercial LiPF_6_ exists. Guo and co-workers added LiPF_6_ to simply transform traditional ether-based 1,3-dioxolane and 1,2-dimethoxyethane to a quasi-solid GPE [163]. They reported various scenarios applying the specialized electrolyte to several different cathode materials. The quasi-solid GPE achieved high stability and high ionic conductivity. This process is versatile; thus, it can be applied to different electrochemical energy storage systems.

Instead of ex-situ methods, several approaches to prepare in situ gelation/polymerization to solve the contact issue between GPE and electrode have been suggested (Figure 18). For example, Song group reported physical gelation of polymer and chemical polymerization of monomer processes and their effect on electrochemical performance [164].

With the same aim of enhancing mechanical properties and ionic conductivity simultaneously, Yang and co-workers synthesized composite electrolytes. They prepared a 3D interconnected porous structure and increased the affinity with liquid electrolytes via compositing bacterial cellulose (BC) and Li_0.33_La_0.557_TiO_3_ nanowires (LLTO NWs) into the aerogel matrix [165] (Figure 19). The affinity provided excellent wettability, and a composite gel electrolyte (CGE) exhibited a Young’s modulus of 1.15 GPa. Furthermore, it exhibited a high ion transference number of 0.88. The robust BC skeleton and LLTO NWs resulted in the suppression of dendritic growth by inducing stable ion deposition.

As another method of fabricating GPEs, the adaptation of ionic liquids (ILs) has been introduced. To realize the uniform Li deposition and effective suppression of dendritic growth, Jin and co-workers researched the immobilization of ILs in GPEs via ion-dipole interactions [166]. To prepare IL immobilized GPEs, they used an imidazolium salt, 1-ethyl-3-methylimidazolium bis(trifluoromethylsulfonyl)-imide (EMI-TFSI) as the IL because of its high ionic strength, electrochemical stability, and nonflammability. Meanwhile, fluorinated copolymers such as PVDF-HFP were used as flexible gel polymer matrices because of their high dielectric constant, high stability, high thermal endurance, and strong mechanical strength. The absorbed IL into copolymer gel electrolyte suppressed dendritic growth by forming a 3D cross-linked network with tethered TFSI anions owing to the strong ion-dipole interactions between imidazolium cations and fluorine atoms. Moreover, it exhibited a self-healing capability for enhanced ionization of Li salts, promoting the Li-ion transport. As the interaction between the imidazolium cations and the C-F dipoles on the polymer chains was strong, a high self-healing capability was obtained. As-made cells with this GPE exhibited a high ionic conductivity, thermal stability, and high mechanical strength, resulting in the effective suppression of dendritic growth during the plating/stripping process.

## 7. Conclusions

In this review, we have summarized the recent progress in functional separators for next-generation batteries. First, existing problems of LMBs are reviewed and various properties that affect battery performance are defined. Second, the types of commercial separators in LIBs and their limitations for LMB applications are described. For the design of stable separators, materials with various compositions and structures have been applied to Li|Li symmetric cells, LMBs, and LSBs. Such various materials can be classified into organic, inorganic, carbon-based, and solid electrolyte materials. We reviewed their properties, synthetic methods, and positive effects in each section. Several remarkable studies were conducted to suppress Li dendritic formation and growth, increase mechanical/thermal/chemical stabilities, maximize the use of active materials, and prevent LiPS shuttle effects, resulting in excellent LMB and LSB performance. The current problems and prospects are categorized according to types of material and summarized below.
Introduction of multifunctional polymer materials into the separators can solve problems such as dendritic growth, poor ionic conductivity, and poor thermal stability. Because only a few types of polymers are applied in separator research, securing new polymers for separator raw materials should be widely conducted. To develop functional separators, polymers should be cheap, easy to prepare, and stable under Li-metal-based battery systems.Inorganic materials can offer robustness to separators by enhancing mechanical properties and minimize formation (or dissolution) of byproducts owing to strong chemical affinity. However, toxic solvents, expensive nanomaterials, or binders are required in most coating methods to functionalize inorganic materials on separators, resulting in environmental or cost concerns. Therefore, researchers should consider eco-friendly methods for inorganic-organic hybrid separators. In addition, for stable LMBs, separators with all-inorganic components could provide excellent stabilities. Because the compositions of inorganic materials are various, we expect the inorganic separators will have significant effects on LMB research.Carbon- or graphene-based composites are commonly used for LSB studies owing to their easy preparation, good conductivity, stability, and good affinity with LiPS. In addition, C-based composites are cost-effective and can form various composites. Therefore, precise control in pore sizes and structures and research on eliminating the risk of direct electron conduction via separator layers are required to develop functional separators.Several studies have focused on applying SE and GPE on LMBs because of their superior mechanical strength. However, compared with the liquid electrolyte system, they exhibit poor electrochemical performance owing to the low conduction at the interface between the separator and electrode. Therefore, reducing the interface resistance is mainly aimed so that they can be used to stabilize LMBs.

As summarized in this review, several researchers have investigated a variety of materials and structures to enhance the performance of separators in LMBs. Therefore, in the near future, we believe functional separators will enable LMBs to be commercialized. We expect this review will provide a general overview and insight into future designs of separators.

## Figures and Tables

**Figure 1 materials-13-04625-f001:**
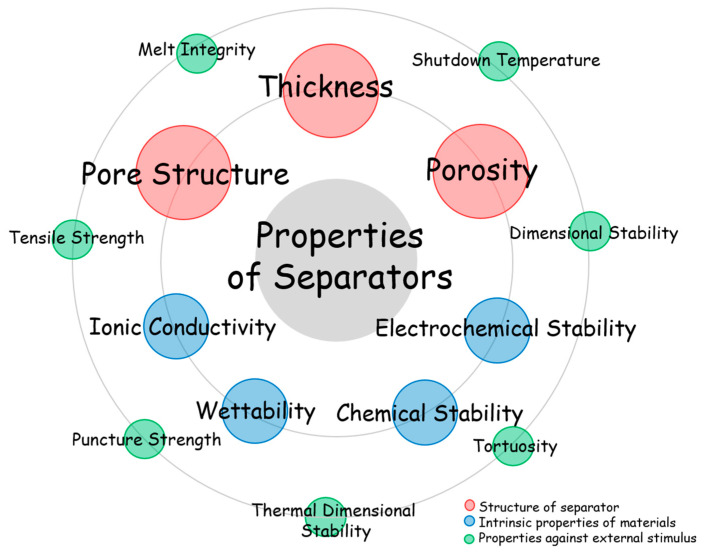
Illustration of the key properties of separators.

**Figure 2 materials-13-04625-f002:**
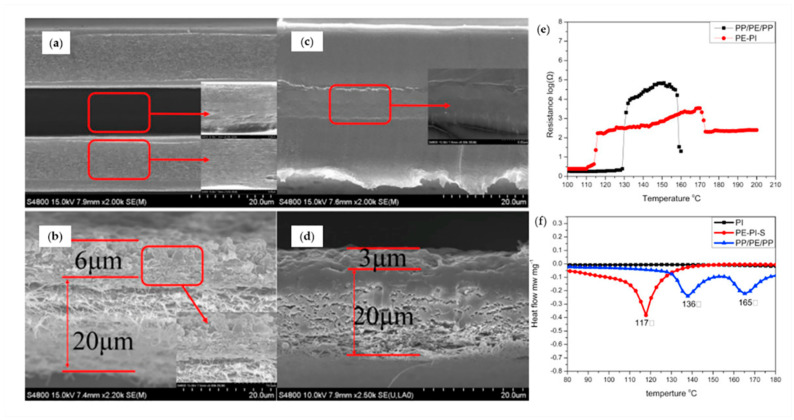
(**a**–**d**) SEM images: (**a**,**c**) cross-section of a PP/PE/PP separator before and after hot treatment at 140 °C for 0.5 h; (**b**,**d**) cross-section of a PE-PI-S nonwoven membrane before and after hot treatment 140 °C for 0.5 h. (**e**) Impedance change with a heating rate 1 °C min^−1^ of the PP/PE/PP separator and PE-PI-S nonwoven membrane. (**f**) differential scanning calorimetry of the PI, PP/PE/PP, and PE/PI/S membranes [48]. Copyright (2015) Elsevier B.V.

**Figure 3 materials-13-04625-f003:**
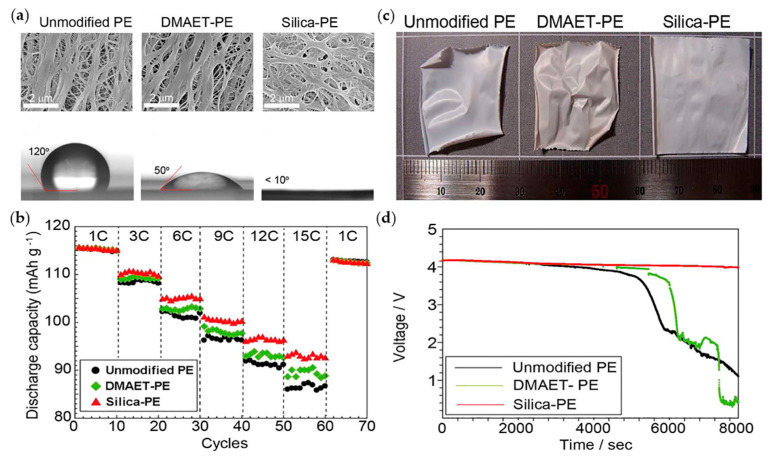
(**a**) SEM and water contact angle images of various membranes. (**b**) Rate capabilities of the different separators (1 C for charge and various C-rates for discharge). The anode and cathode are graphite and LiMn_2_O_4_, respectively. (**c**) Digital images of the different separators after exposure to 140 °C for 1 h. (**d**) Open circuit voltage measurements of the cells with various separators at 140 °C. Reprinted with permission from [51]. Copyright (2012) American Chemical Society.

**Figure 4 materials-13-04625-f004:**
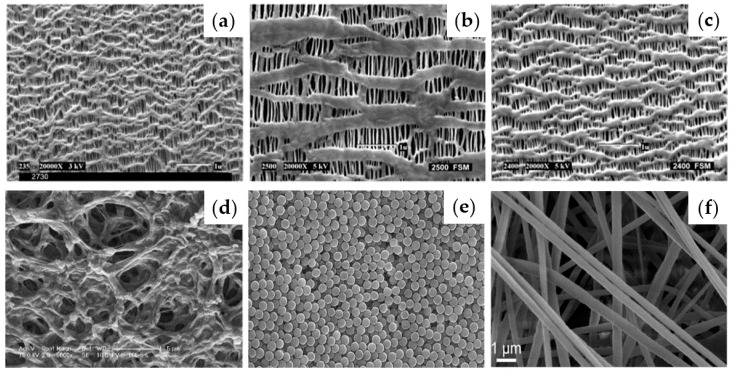
SEM images of the surfaces of various separators: (**a**) Celgard 2730 (PE), (**b**) Celgard 2500 (PP), (**c**) Celgard 2400 (PP). Reprinted with permission from [41]. Copyright (2004) American Chemical Society. (**d**) PVDF membrane (ethanol precipitation bath). Reprinted with permission from [57]. Copyright (2008) Elsevier B.V. (**e**) PMMA particles Reprinted with permission from [58] Copyright (2018) CC BY. (**f**) PAN nanofibers Reprinted with permission from [59] Copyright (2019) CC BY.

**Figure 5 materials-13-04625-f005:**
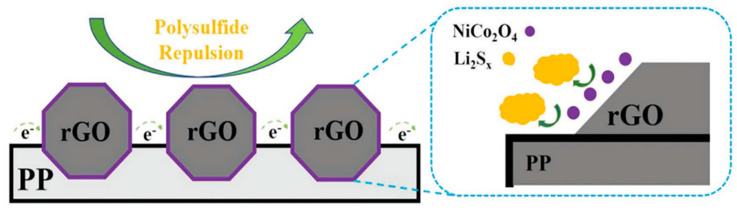
Schematic of the NiCo_2_O_4_@rGO/PP when recharging. Reprinted with permission from [83]. Copyright (2019) WILEY-VCH.

**Figure 6 materials-13-04625-f006:**
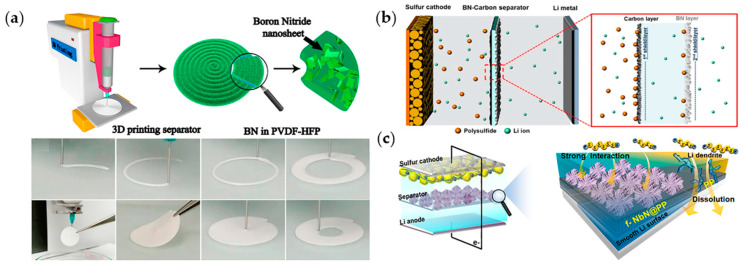
(**a**) Schematic of the 3D printing apparatus, the BN in the PVDF-HFP separator, and the corresponding composition, and digital photographs at different points in the printing process. Reprinted with permission from [96]. Copyright (2017) Elsevier B.V. (**b**) Schematic of the systemic role of a BN-carbon separator in the discharge process. Reprinted with permission from [97]. Copyright (2017) Springer Nature. (**c**) Schematic of the effect of the NbN@PP separator. Reprinted with permission from [98]. Copyright (2020) American Chemical Society.

**Figure 7 materials-13-04625-f007:**
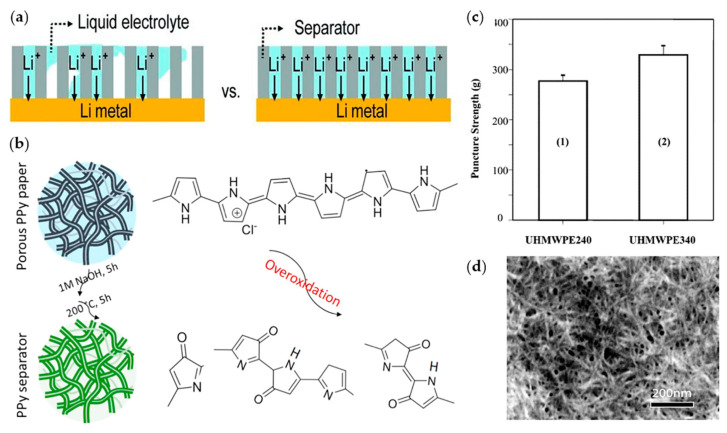
(**a**) Comparison of poor wettability (left) and good wettability (right). Reprinted with permission from [102]. Copyright (2012) WILEY-VCH. (**b**) Schematic of the fabrication of the highly overoxidized PPy paper membrane using heat treatments and sequential base. Overoxidation changes the structure of PPy. Reprinted with permission from [104]. Copyright (2018) Elsevier B.V. (**c**) Puncture strength according to polymer ratio (HDPE:UHMWPE = 27:3); (1) UHMWPE, Mw = 240,000,000 and (2) UHMWPE, Mw = 340,000,000. Reprinted with permission from [105]. Copyright (2002) Elsevier Science B.V. (**d**) SEM images of a PBO-NF membrane. Reprinted with permission from [106]. Copyright (2016) American Chemical Society.

**Figure 8 materials-13-04625-f008:**
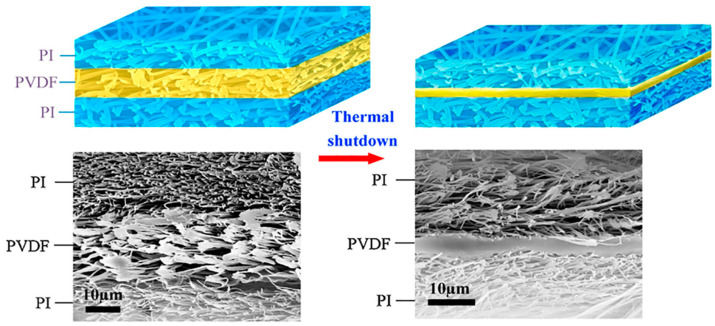
Figure and cross-sectional SEM images of the PI/PVDF/PI membrane before and after the hot melting and shutdown behavior of the PVDF nanofibers layer. Reprinted with permission from [115]. Copyright (2015) Elsevier Ltd.

**Figure 9 materials-13-04625-f009:**
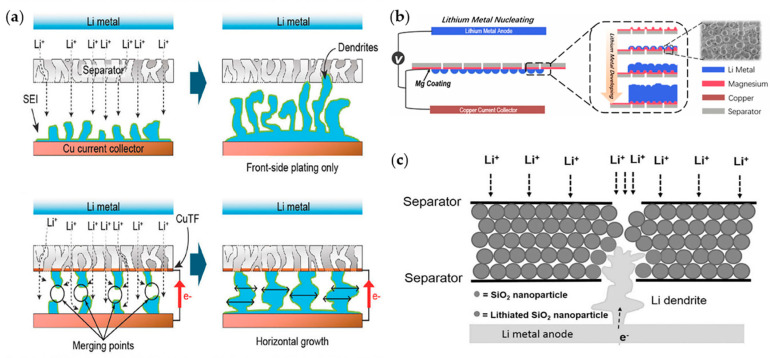
(**a**) Schematic of the Li-metal deposition mechanism using the bare PE separator (top) and PE/CuTF Janus separator (bottom). Reprinted with permission from [82]. Copyright (2017) WILEY-VCH. (**b**) Schematic of electrodeposition on a Mg-coated separator. Reprinted with permission from [81]. Copyright (2018) Elsevier B.V. (**c**) Schematic of the mechanism with an extended battery life with a silica nanoparticle sandwiched separator. Reprinted with permission from [118]. Copyright (2016) WILEY-VCH.

**Figure 10 materials-13-04625-f010:**
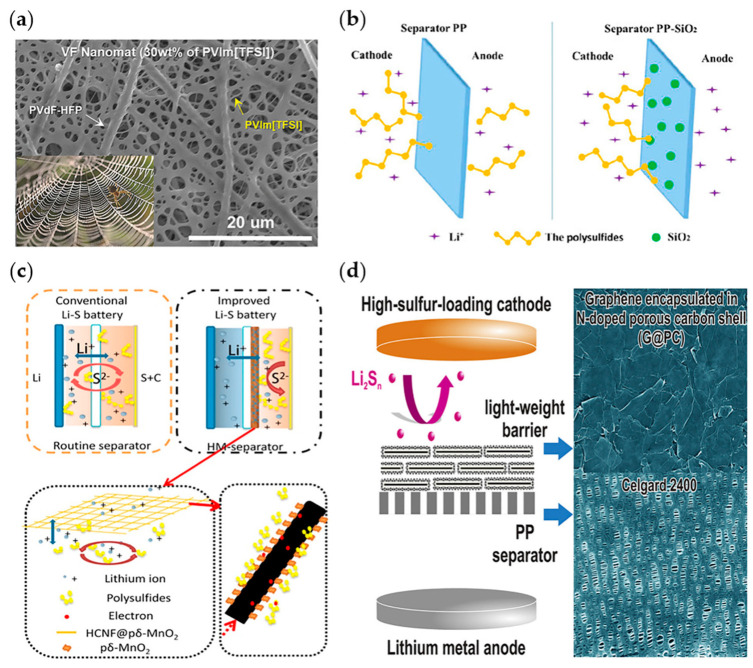
(**a**) SEM image of the VF nanomat (30 wt.% of PVIm[TFSI]), well-developed porous structure. Reprinted with permission from [127]. Copyright (2018) WILEY-VCH. (**b**) Schematic of the mechanism of suppressing shuttle effects. Left and right for PP separator and PP-SiO_2_ separator, respectively. Reprinted with permission from [84]. Copyright (2017) American Chemical Society. (**c**) Schematic of battery configurations with the routine and fabricated separators, and the function mechanism of the fabricated separator. Reprinted with permission from [129]. Copyright (2017) Elsevier B.V. (**d**) LiPS blocking process and SEM images of nitrogen-doped carbon-coated on PP membrane. Reprinted with permission from [130]. Copyright (2018) Elsevier Inc.

**Figure 11 materials-13-04625-f011:**
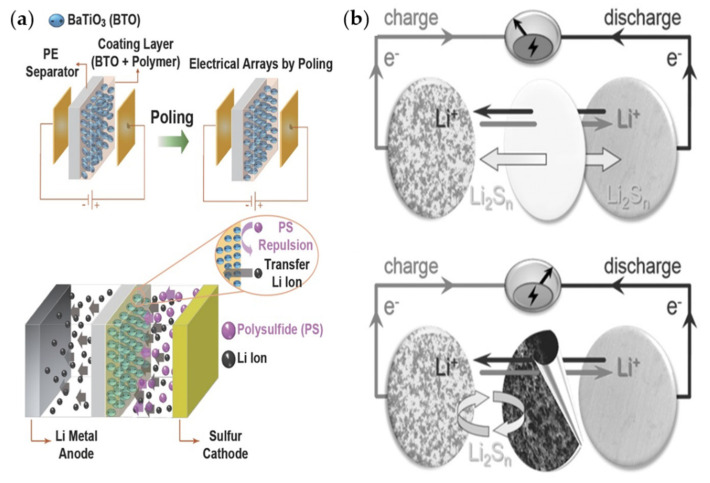
(**a**) Schematic of the poling process of the BTO-coated PE separator and the effect of polysulfide rejection. Reprinted with permission from [139]. Copyright (2016) WILEY-VCH. (**b**) Schematic configuration of the Li–S cells with a commercial separator (top) and mesoporous carbon-coated separator (bottom). The cell configuration consists of, from left to right, a sulfur cathode, the corresponding separator, and a Li-metal anode. Reprinted with permission from [140]. Copyright (2015) WILEY-VCH.

**Figure 12 materials-13-04625-f012:**
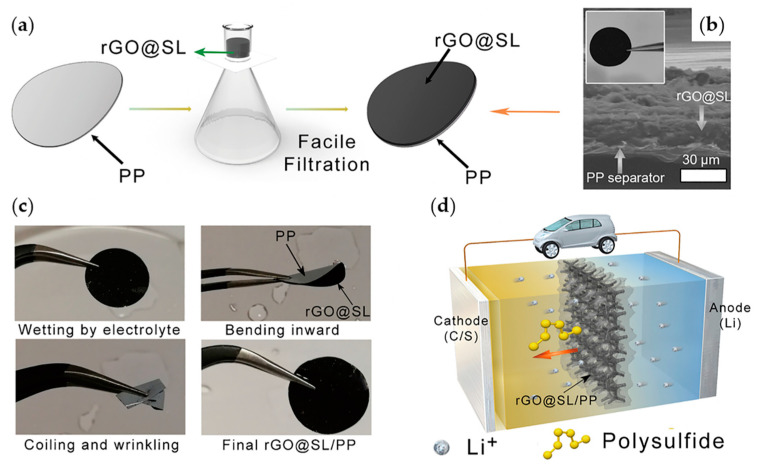
(**a**) Schematic of the fabrication procedure of the rGO@SL/PP separator (**b**) SEM image of the rGO@SL/PP separator (**c**) Digital photos of the rGO@SL/PP separator under various mechanical stresses. (**d**) Schematic of the rGO@SL/PP separators inhibiting shuttle effects in LSBs. Reprinted with permission from [141]. Copyright (2018) Elsevier Inc.

**Figure 13 materials-13-04625-f013:**
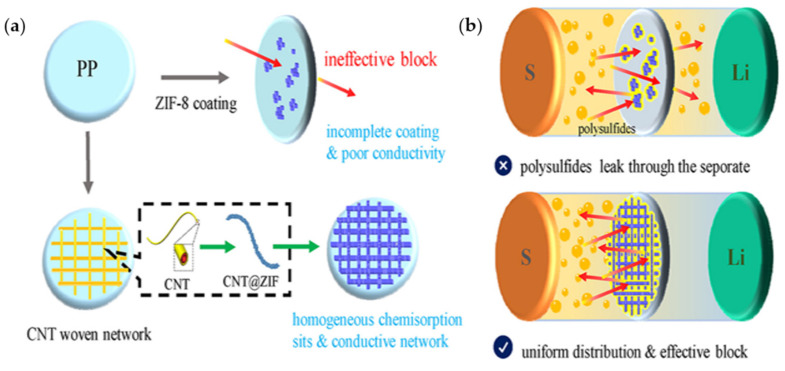
(**a**) Schematics of ZIF-8 and the CNT@ZIF composite modified separators (**b**) Li-S battery configuration with the modified separators. Reprinted with permission from [142]. Copyright (2018) Elsevier B.V.

**Figure 14 materials-13-04625-f014:**
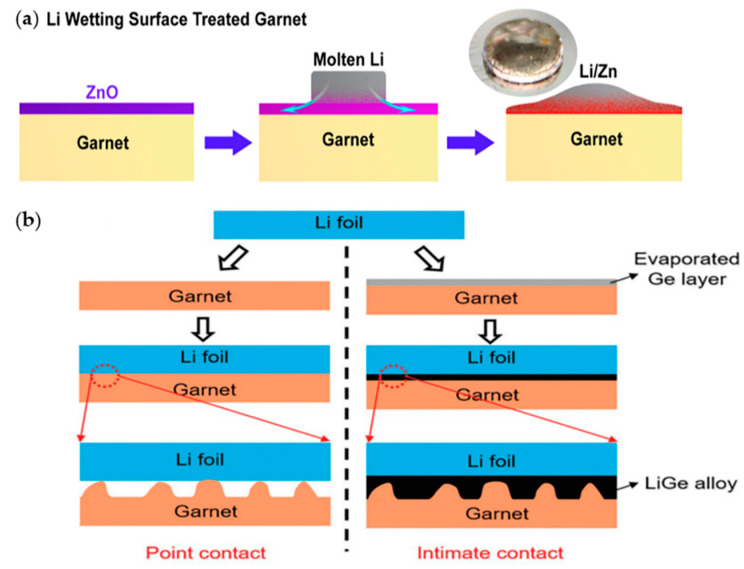
(**a**) ZnO-coated garnet electrolyte. Reprinted with permission from [153]. Copyright (2017) American Chemical Society.; (**b**) Ge-coated garnet electrolyte. Reprinted with permission from [154]. Copyright (2017) John Wiley and Sons.

**Figure 15 materials-13-04625-f015:**
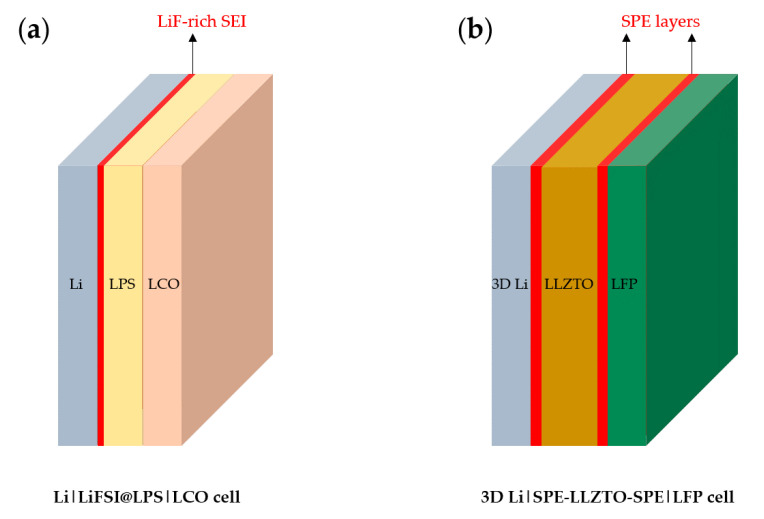
Electrolyte interphase layers between electrolytes and electrodes. (**a**) Scheme of Li|LiFSI@LPS|LCO cell. (**b**) Scheme of 3D Li|SPE-LLZTO-SPE|LFP cell.

**Figure 16 materials-13-04625-f016:**
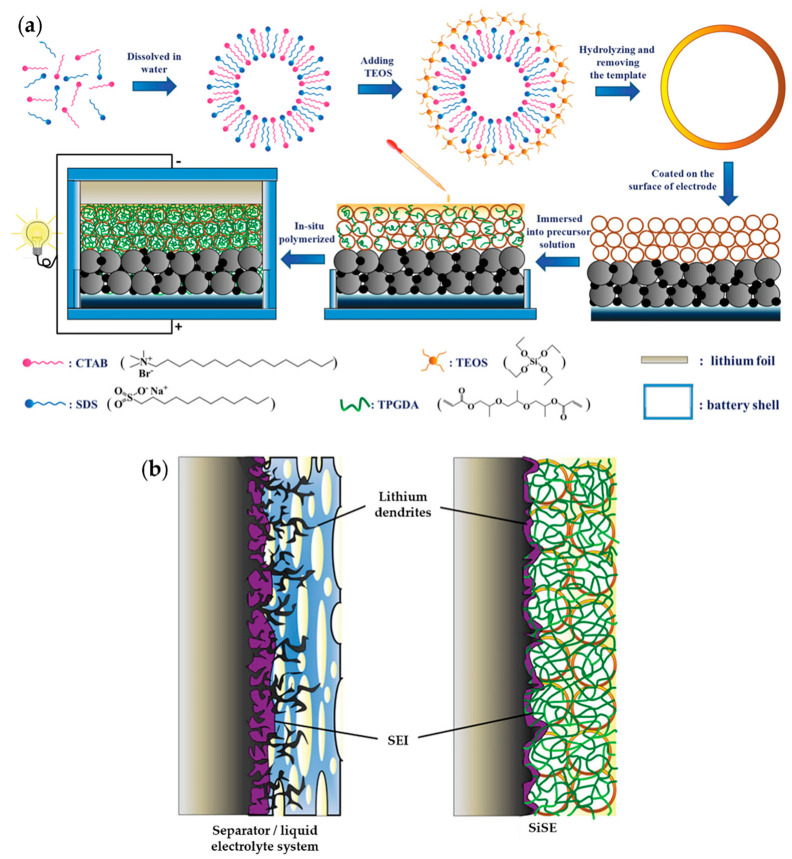
(**a**) Illustration for preparation steps of SiSE. (**b**) Comparison of SiSE to separator/liquid electrolyte system. Reprinted with permission from [161]. Copyright (2016) John Wiley and Sons.

**Figure 17 materials-13-04625-f017:**
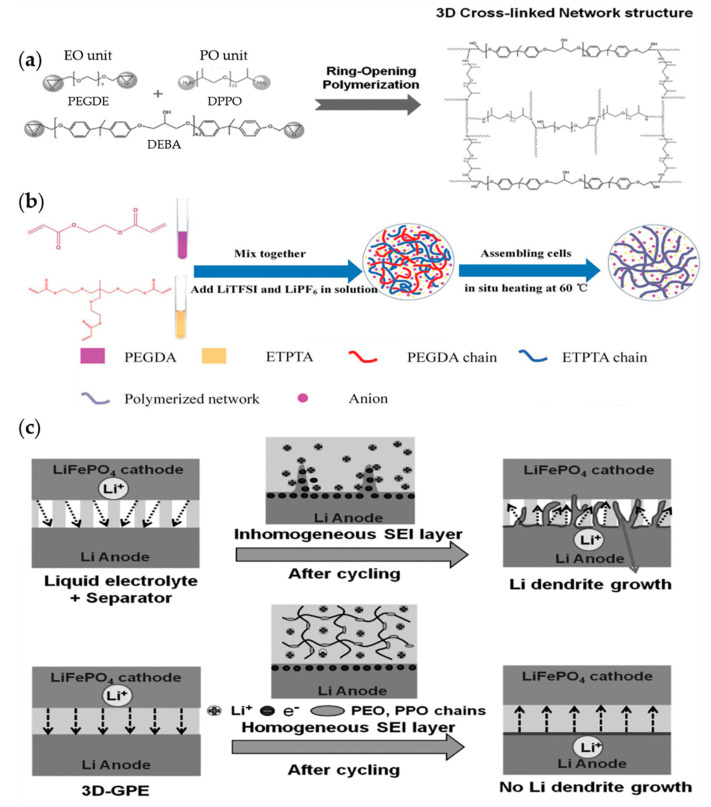
(**a**) Illustration of the synthesis of the GPE. Reprinted with permission from [148]. Copyright (2017) John Wiley and Sons. (**b**) Step process for in situ polymerization of GPE. Reprinted with permission from [162]. Copyright (2018) John Wiley and Sons. (**c**) Schematics of the changes in the Li electrodes using a liquid electrolyte and 3D-GPE during the Li plating/stripping. Reprinted with permission from [148]. Copyright (2017) John Wiley and Sons.

**Figure 18 materials-13-04625-f018:**
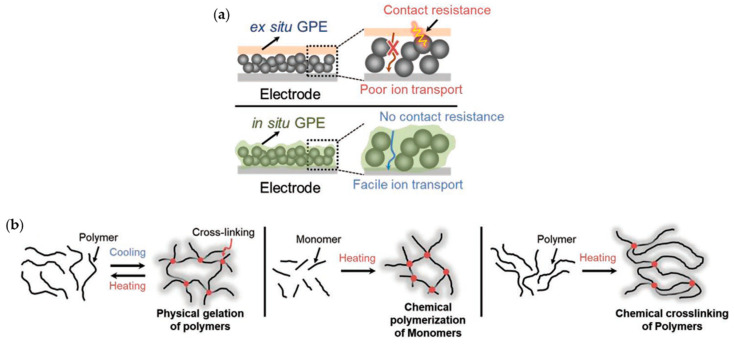
(**a**) Ex-situ GPE and in situ GPE. (**b**) Gelation via cross-linking of polymers. Reprinted with permission from [164]. Copyright (2018) John Wiley and Sons.

**Figure 19 materials-13-04625-f019:**
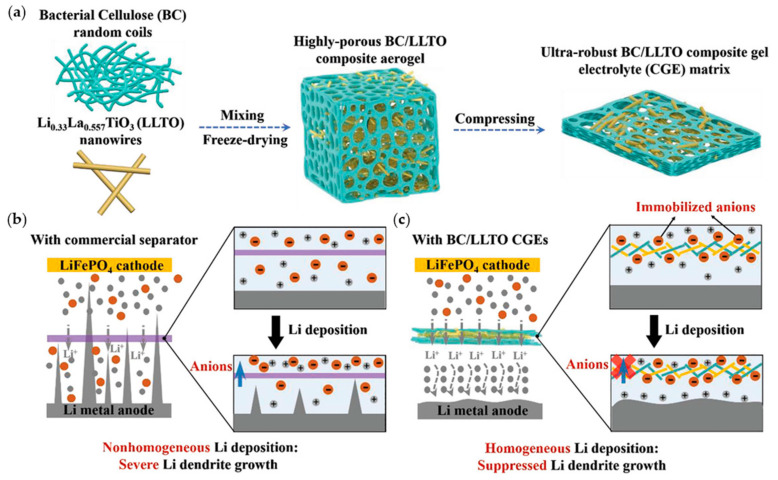
Preparation of BC/LLTO CGEs. (**a**) Schematic of the effect of (**b**) a commercial separator and (**c**) a BC/LLTO CGE on Li deposition in a battery. Reprinted with permission from [165]. Copyright (2019) WILEY-VCH.

**Table 1 materials-13-04625-t001:** Gurley value, ionic conductivity, MacMullin of a PP/PE/PP separator and CNT separators as a function of the IPA-water composition ratio [46]. Copyright (2012) ROYAL SOCIETY OF CHEMISTRY.

IPA-Water Ratio (vol/vol%)	Gurley Value [s 100cm^−3^ Air]	Ionic Conductivity [mS cm^−1^]	MacMullin Number
60/40	Too high to be determined	0.02	337.6
80/20	850	0.53	14.1
95/5	496	0.75	10.1
100/0	487	0.77	9.9
PP/PE/PP separator	500	0.73	10.3

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
