# Peer review of "A Review of Functional Separators for Lithium Metal Battery Applications"

_materials, 2020, doi:10.3390/ma13204625_

Round 1

Reviewer 1 Report

Present manuscript describes review on current state-of-art in manufacture of functional separators for Li-metal batteries. The investigation of improvement safety and efficiency of the Li-metal batteries is hot topic and rapidly evolving research area. However, article is suffered from poorly conducted comparison with latest literature data. The language of the manuscript is clear and understandable, however the logic of composition of the review is quite complicated.

There are some concerns that need to be addressed:

  • The major drawback of current study that not all previously reported reviews and research papers on functional separators are taken into consideration, e.g.:
  1. 10.1093/nsr/nwx037
  2. 10.1016/j.jpowsour.2016.09.044
  3. 10.1021/acsami.8b14196
  • I strongly recommend sorting separators not only based on its materials – but also by their function or properties – as it was stated in the introduction. It can be grouped like dendrite-suppressing separators, thermostable separators, separators from sustainable materials or earth-abundant elements, etc.
  • Figure 1 the figure is nice, however does not carry a deep meaning and the colour codes is not clear.
  • 3 the μm units are poorly visible.
  • 4 lines 105-104 – lack of the necessary citation.

Author Response

We have attached a letter to Reviewer 1. Please see the attachment.

Reviewer 2 Report

The review is well written and timely, however the Authors ought to

consider earlier reviews on the topic, e.g.

Functional membrane separators for next-generation
high-energy rechargeable batteries
Yuede Pan, Shulei Chou, Hua Kun Liu and Shi Xue Dou, National Science Review
4: 917–933, 2017

as well as relevant papers, e.g. 

Kim, J.Y., Shin, D.O., Kim, K.M. et al. Graphene Oxide Induced Surface Modification for Functional Separators in Lithium Secondary Batteries. Sci Rep 9, 2464 (2019)

BinLiu, Ji-GuangZhang,

Volume 2, Issue 5, 16 May 2018, Pages 833-845   This activity by the Authors will make the review more comprehensive and updated

Author Response

We attached a letter to reviewer file. Please find the attachment. :)

Reviewer 3 Report

The authors have produced a fairly wide-ranging review of lithium battery separators with a lot of topics that are good to bring up: conductivities, mechanical strength, and nontoxicity. My major critique is that the order in which this information is presented is seemingly random, with the major flaw being the entirety of section 4 before section 5. In its current form the authors treat lithium/sulfur batteries especially different from the remainder of lithium ion/lithium metal batteries and do so with no introduction. Hence, properties which are viable in LIB separators become magnified or muddled in applications of Lithium/sulfur batteries because of new shuttle effects, polysulfide dissolution, etc.

Effectively, the review reads as 1. The properties of a great LIB separator; 2. Examples on how research has made better lithium/sulfur batteries (which need other properties, not mentioned in part 1), 3. Examples on how research has made better batteries with lithium metal; 4. Examples on better lithium/sulfur separators. A rearrangement of this content is in order, as otherwise the review reads as highly scattered.

Some specific remarks below:

L167, Sec 2.7: How is shrinkage measured? The article cited by the authors in this section is better if expanded a bit more: e.g., "prevented shrinkage of the entire separator"- due to the silica component? "Improved rate performance" and "improved thermal stability" over what? Ideally the reference in this section is one in which shrinkage is quantified and direct metrics compared.

L199 "inherent properties....restrict the performance of the batteries" and L 204: "may degrade battery performance"- In what way? Lower capacity? Rate? The dendrite problem is often a rate problem. Is this not a tradeoff with cyclability? The authors do a good job in L212-214 highlighting how the alternatives are limited, but not how the current ones are limited.

L236: This appears to be the first instance where authors use both acronyms LSBs and LiPS, and these are not defined prior to this. I assume this means sulfur batteries and lithium polysulfides? This is an immensely abrupt segue as the manuscript up to this point discusses lithium ion batteries as a whole and there is no mention in the first pages about how polysulfides interact with the separator. This profound shift is not good for the reader. The authors introduce the importance of mechanical stability of separators, thermal stabilities, etc....then when getting into actual examples they discuss LiPS absorption. Where was this in the necessary properties of a separator section? The reader finally finds these strategies of mechanical properties in lines 406-433.

L294: "Carbon alone cannot be used as a separator because it has high electrical conductivity." Immediately followed by "Graphene has excellent conductivity....it is commonly used to improve the performance of separators." What? What happened to "The separator must be an electronic insulator" on L138? And then L324 "BN is a crucial material because of its electrically insulating properties"

L521: Separators made from nontoxic and sustainable.....

Reviewer 4 Report

The manuscript is a review of recent advances in the field of Lithium metal batteries. This hot and promising topic is changing very fast and therefore it is important for researchers to know the latest achievements. In my opinion, such a format should be supported, since the authors, being qualified in the field, performed a critical analysis of recent developments, and showed the main research directions.

The article is well structured and is of interest to a wide range of readers.

Minimal comments/remarks that I had when reading the text are more likely to taste, so I think the article can be published in its current form.

Author Response

Please check the attached file :)

Round 2

Reviewer 1 Report

Revised version of the manuscript is much more clear and logical. Readability has improved significantly. 

Please check that you have received permission from the publishers for all graphics used in this review (e.g. Figure 15).

Author Response

Please check the attached file :)

Reviewer 3 Report

My comments were addressed satisfactorily.